

# The CO5 configuration of the 7km Atlantic Margin Model: Large scale biases and sensitivity to forcing, physics options and vertical resolution

Enda O'Dea[1], Rachel Furner[1], Sarah Wakelin[2], John Siddorn[1], James While[1], Peter Sykes[1], Robert King[1], Jason Holt[2], and Helene Hewitt[1]

[1] Met Office, Exeter, UK
[2] National Oceanography Centre, Liverpool, UK

*Correspondence to:* Enda O'Dea (enda.odea@metoffice.gov.uk)

**Abstract.**

   We describe the physical model component of the standard Coastal Ocean version 5 configuration (CO5) of the European North West Shelf (NWS). CO5 was developed jointly between the Met Office and the National Oceanography Centre. CO5 is designed with the seamless approach in mind, which allows for modeling of multiple timescales for a variety of applications

from short-range ocean forecasting through to climate projections. The configuration constitutes the basis of the latest update to the ocean and data assimilation components of the Met Office's operational Forecast Ocean Assimilation Model (FOAM) for the NWS. A 30.5 year non-assimilating control hindcast of CO5 was integrated from January 1981 to June 2012. Sensitivity simulations were conducted with reference to the control run. The control run is compared against a previous non-assimilating Proudman Oceanographic Laboratory Coastal Ocean Modelling System (POLCOMS) hindcast of the NWS. The CO5 control

hindcast is shown to have much reduced biases compared to POLCOMS. Emphasis in the system description is weighted to updates in CO5 over previous versions. Updates include an increase in vertical resolution, a new vertical coordinate stretching function, the replacement of climatological riverine sources with the pan-European hydrological model E-HYPE, a new Baltic boundary condition and switching from directly imposed atmospheric model boundary fluxes to calculating the fluxes within the model using bulk formula. Sensitivity tests of the updates are detailed with a view to attributing observed changes in the

new system from the previous system and suggesting future directions of research to further improve the system.

## 1   Introduction

The European North West Shelf (NWS) is an area of intense socioeconomic interest with a wide variety of dynamical regimes. It is a region that has been the subject of numerous research models over many years both domain wide and focusing on smaller subregions. Research models and associated assimilation schemes for the region have matured into a number of operational

systems. As part of the Copernicus Marine Environment Monitoring Service (CMEMS), an operational forecast system based on the Atlantic Margins Model (AMM) (O'Dea et al., 2012) has been developed to provide products for coastal modeling downstream users. The AMM domain for the NWS is shown in Fig. 1.



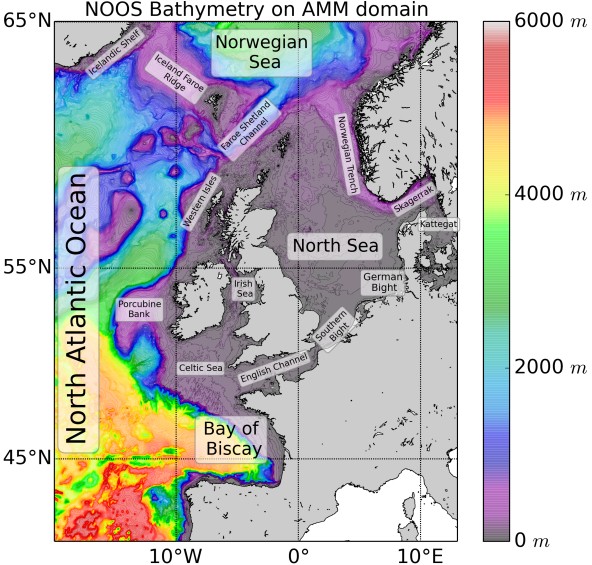

**Figure 1.** NOOS bathymetry for the AMM domain

In compliment to the forecast systems, CMEMS also make reanalysis products available to the end users. The reanalysis products not only provide end users with data from past decades, but also provide a way to assess and validate the operational systems over longer periods against historical data. The presentation of systematic biases and drifts allows the users to understand the limitations and appropriateness of a particular product to their interest or application. Furthermore, as systems are upgraded the associated reanalyses provide a means to inter-compare and evaluate the effectiveness of system updates.

In this paper the subject of interest is what we label the standard Coastal Ocean configuration version 5 (CO5). CO5 includes all input parameters, ancillary files, model code and compilation keys required to run the model. CO5 forms the physics component of the Copernicus reanalysis product replacing the preexisting POLCOMS derived hindcast product. In support of the full reanalysis a non-assimilative control hindcast was integrated from January 1981 to June 2012. CO5 was jointly developed by the Met Office and the National Oceanography Centre. Standard configurations such as CO5 are subsequently incorporated as constituent parts of broader modeling systems such as climate projections (Tinker et al., 2015) or coupled systems (Sikirić et al., 2013).

CO5 is an update of the Nucleus for European Modelling of the Ocean (NEMO) (Madec, 2008) configuration used to model the NWS in O'Dea et al. (2012). For convenience we reference the configuration in O'Dea et al. (2012) as CO4. Changes include new riverine forcing, updated Baltic boundary conditions, increased vertical resolution as well as updating the base NEMO version from 3.2 to 3.4. Here we describe the non-assimilating CO5 control hindcast that provides a reference to understand underlying biases and drifts attributable to changes in the physics updates alone. The CO5 reanalysis product is an update to the 12 km POLCOMS hindcast (Holt et al., 2012) for 1965-2004 of the same region. We compare the CO5 non-





assimilative control hindcast with the POLCOMS hindcast over common years of integration 1985-2004 and exclude the CO5 spin-up years 1981-1984. Both are compared against standard climatologies and observations. Individual updates incorporated into CO5 are also investigated systematically by a series of 30 year sensitivity experiments, looking at the changes in isolation. The surface and boundary forcing datasets used in CO4 only start from 2006 so it is not possible to do a full 30 year like for like

CO4 and CO5 intercomparison. However, shorter 5 year experiments looking at the effects of the forcing are also investigated.

The structure of the paper is as follows. Section 2 gives an overview of the standard configuration CO5. Configuration updates are detailed in Section 3. The experimental design including the specifics of the sensitivity experiments are outlined in Section 4. Section 5 has three main subsections:

- 5.1 is concerned with tidal analysis of CO5.

- 5.2 isolates long term biases compared to climatology, observations and the POLCOMS hindcast.

- 5.3 presents results from sensitivity experiments that look in isolation at changes brought into CO5.

Section 6 summarizes and discusses the results before commenting on future system upgrades which are informed by the analysis of this paper.

## 2 Core Model description

CO5 builds upon and thus shares many of the core features of the previous Met Office shelf seas model configuration CO4, as described in O'Dea et al. (2012). Elaboration of the key features particular to CO5 that are distinct from CO4 is deferred to Section 3.

CO5 is based on version 3.4 of NEMO (Madec, 2008). The model domain extends from $(20°$ W, $40°$ N$)$ to $(13°$ E, $65°$ N$)$ on a regular latitude-longitude grid. The domain covers the entirety of the European North West Shelf and includes a sufficient

portion of the deep waters of the eastern North Atlantic to encapsulate cross shelf break exchange. The bathymetry for CO5 is derived from the North-West Shelf Operational Oceanographic System (NOOS) bathymetry. The NOOS bathymetry is a combination of GEBCO 1' data and a variety of local data sources from the NOOS partners. The meridional grid resolution is $1/15°$ or $7.4\,\mathrm{km}$. The zonal resolution of $1/9°$ varies from $9.4\,\mathrm{km}$ along the southern boundary to $5.2\,\mathrm{km}$ along the northern boundary with a mean of $7.4\,\mathrm{km}$ at $52.5°$ N. Although the grid horizontal resolution readily resolves the external Rossby radius

$(200\,\mathrm{km})$, it is not sufficient to resolve the internal Rossby radius on the shelf which is of order $4\,\mathrm{km}$ (Holt and Proctor, 2008). However, at the time of integration of the reanalysis, it was not computationally feasible to conduct multiple 30 year hindcasts of the CO5 domain with a resolution approaching the $1.5\,\mathrm{km}$ required to resolve the internal radius.

As tides and surges play such important roles on the European North West Shelf, a non linear free surface is implemented using the variable volume layer (Levier et al., 2007) and time splitting approaches in NEMO. The baroclinic time step used

in the 30 year hindcasts of CO5 is 300 seconds with a barotropic time step of 10 seconds. The advection of momentum is both energy and enstrophy conserving (Arakawa and Lamb, 1981). Both bi-Laplacian and Laplacian horizontal viscosities are applied. The Laplacian viscosity is applied along geopotential levels with a coefficient of $30.0\,\mathrm{m^2\,s^{-1}}$. The bi-Laplacian





viscosity is used to retain model stability and is applied on model levels with a coefficient of $1.0 \times 10^{-10}\,\mathrm{m^4\,s^{-1}}$. The lateral momentum boundary condition is free slip. Tracer advection is implemented using the total variation diminishing (TVD) scheme (Zalesak, 1979). Unlike CO4, Laplacian tracer diffusion operates only along geopotential levels with a coefficient of $50\,\mathrm{m^2\,s^{-1}}$.

The Generic Length Scale (GLS) turbulence scheme calculates the turbulent viscosities and diffusivities (Umlauf and Burchard, 2003). The second-moment algebraic closure of Canuto et al. (2001) is solved with two dynamical equations (Rodi, 1987) for the turbulence kinetic energy (TKE), $k$ and TKE dissipation, $\epsilon$ (Umlauf and Burchard, 2005). At the surface and bed, Neumann boundary conditions on $k$ and $\epsilon$ are applied. Surface wave mixing is parameterized as in Craig and Banner (1994). Dissipation under stable stratification is limited using the Galperin limit (Galperin et al., 1988) of 0.267. A spatially varying

log layer derived drag coefficient with a minimum set at 0.0025 controls the bottom friction.

## 3   Summary of main model updates

CO5 has four configuration updates from CO4. These updates involve the vertical levels, the source riverine input, the treatment of the exchange with the Baltic through the Kattegat and the base version of NEMO. Furthermore, the inputs at the oceanic lateral boundary conditions and the surface boundary condition for the 30 year hindcast are substantially different from the

shorter runs detailed for the forecast implementation of CO4 in O'Dea et al. (2012). Here we describe in detail each of the changes and in Section 4 a set of sensitivity experiments explores the impacts of these changes.

    Relative to CO4, which uses the stretching function in Song and Haidvogel (1994), CO5 features both more model levels (increased from 33 to 51) and uses the stretching function as detailed in Siddorn and Furner (2013) for the terrain following coordinate system. We refer to the stretching function in CO4 as SH and that in CO5 as SF. The new stretching function

maintains near uniform vertical resolution at the surface. Keeping the surface vertical resolution almost the same across most of the domain implies a more consistent air sea exchange domain wide. The stretching function also aims to minimize horizontal pressure gradient errors induced by sloping horizontal model levels. A comparison of the thickness of the surface model level in CO4 and CO5 is shown in Figure 2. It is only in the shallowest regions (bathymetry $< 50\,\mathrm{m}$) where the surface level thickness in CO5 is not set equal to $1\,\mathrm{m}$, whereas in CO4 the surface model level varies considerably over the domain from deep water to

shelf. Thus, any change in CO5 that impacts upon air-sea exchange will be applied equally across most of the domain allowing cause and effect to be more readily parsed. Furthermore, follow on configurations of CO5 will feature ocean-atmosphere coupling where again consistent air-sea exchange will be important.

    The second significant change between CO4 and CO5 is the data source for riverine input. In CO4 an annual climatology of some 320 European rivers mapped to 165 outflow points on the CO4 grid constitutes the riverine input regardless of the

model year (Young and Holt, 2007). As a step towards temporal variation and higher resolution of riverine sources the old climatology is replaced with data from a pan-European implementation of the hydrological model HYdrological Predictions for the Environment (HYPE) (Lindström et al., 2010). The European implementation of HYPE is known as E-HYPE (Donnelly et al., 2015) and has a sub-basin resolution of $120\,\mathrm{km^2}$. There is both an operational forecast and hindcast of E-HYPE and the





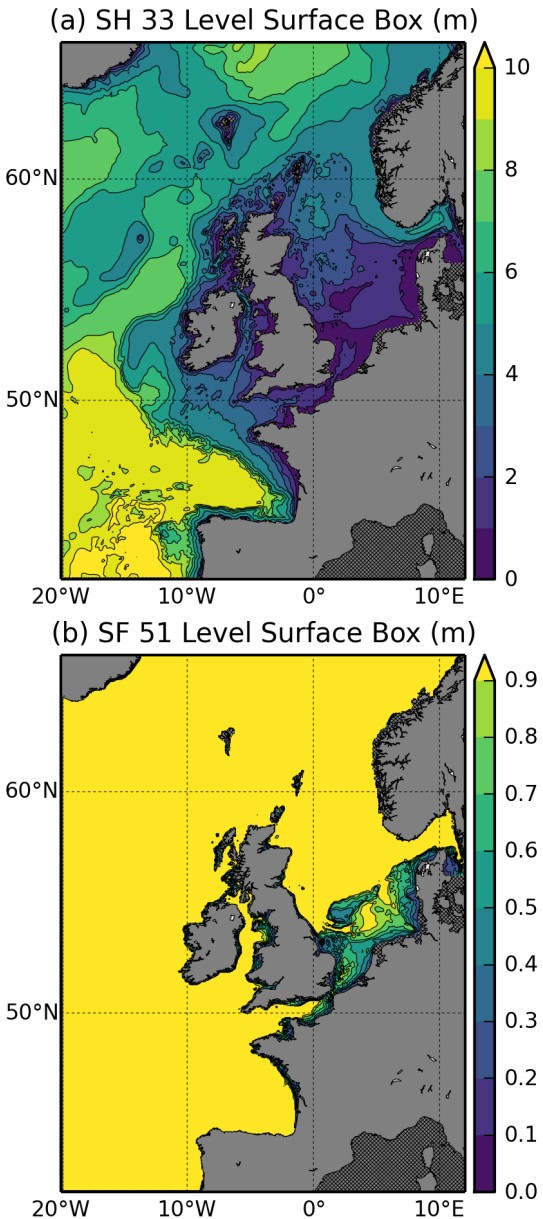

**Figure 2.** Thickness of surface model levels in CO4 (a) and CO5 (b).

data is freely available. Daily river outflow data is mapped to 476 outflow points on the CO5 grid from version 2.1 of E-HYPE. The data was provided by the Swedish Meteorological and Hydrological Institute (SMHI) for the entire period of the hindcast. The E-HYPE data provides a greater number of river sources along the coastline of continental Europe. Figure 3 compares the total riverine input from all rivers in the domain for both the CO4 river climatology and the 1980-2012 mean of the E-HYPE



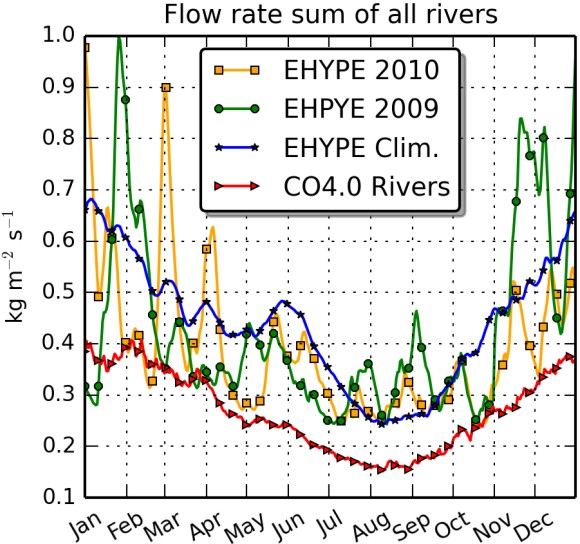

**Figure 3.** Comparison of total river flow rate between E-HYPE individual years, 30 year mean and CO4 climatological rivers

data. Two individual years of E-HYPE data are also included to show the day to day and year to year variability that E-HYPE daily data contains compared to the climatological means. The increase in continental river outflow leads to the mean E-HYPE outflow being considerably larger than the CO4 river climatology. However, the increase is not uniform and indeed the mean outflow from the British Isles is actually slightly reduced in E-HYPE. In some areas such as the German Bight the E-HYPE

outflow is substantially increased.

The third update to CO5 concerns the exchange between the North Sea and the Baltic through the Danish straits and the Kattegat. At 7 km resolution it is not possible to resolve the Danish straits, given that Öresund is 4 km wide at its narrowest. Thus, alternative approaches are required. The approach in CO4 was to apply a daily climatological flux through two additional river points at roughly where the Great Belt and the Öresund open to the Kattegat. If the flux is negative, that is water leaves

the Kattegat and enters the Baltic, ocean water is removed at the river point according to the magnitude of the flux. If the flux is positive, a flux of water of specified salinity and temperature is added at the river point. In CO5 a different approach is taken and involves the specification of a new lateral boundary condition with a relaxation zone spread across the Kattegat. No attempt is made to model the Danish straits and they are removed from the domain. The data for the lateral boundary condition comes from a General Estuarine Transport Model (GETM) of the North Sea and the Baltic Sea. The North Sea-Baltic Sea (NSBS)

model was run at the Leibniz-Institut für Ostseeforschung Warnemünde (IOW) (Gräwe et al., 2015). The horizontal resolution was 1 nautical mile and there are 50 vertical levels. The version of GETM was v2.3.1. Daily NSBS data is only available from 2001-2012 and a climatology of this daily boundary data is created to cover 1981-2001. Temperature and salinity data are relaxed over the relaxation zone. Barotropic velocity and sea surface elevation boundaries from the NSBS model can also be prescribed by the Flather radiation boundary condition. However, the reference elevation in the NSBS model and the data from



the models of the Atlantic into which CO5 is nested are not the same. Such a difference could lead to a persistent flux in or out of the Baltic that is not physically based. An anomaly of elevation about a mean value at the boundary could provide a suitable solution. For the hindcast we describe here, only relaxation of the temperature and salinity is used, though a sensitivity run including elevation was conducted.

The surface boundary condition in CO5 has also changed from CO4. In CO4 the surface boundary conditions are directly prescribed fluxes from the Met Office's Numerical Weather Prediction (NWP) model. Directly prescribed fluxes are replaced by calculating momentum, heat and freshwater fluxes using the Common Ocean-ice Reference Experiment (CORE) bulk formulae (Large and Yeager, 2009). The atmospheric forcing dataset used to force the 30 year hindcast is the ERA-Interim dataset of the European Centre for Medium-Range Weather Forecasts (ECMWF) (Dee et al., 2011). In addition to switching to bulk formulae

the light attenuation scheme used in CO5 is also changed to the standard NEMO tri-band Red-Blue-Green (RGB) scheme of Lengaigne et al. (2007). The RGB scheme replaces the single band scheme presented in Holt and James (2001) which is used in CO4. We refer to this single band scheme as PDWL in this paper. One consequence of this change in light scheme in CO5 is that the extinction depths do not vary across the domain in proportion to the bathymetry as in CO4 and POLCOMS. The variance in extinction depth was a first order attempt to mimic the change in water clarity from deep waters to shallow.

## 15   4   Experimental Design

The CO5 control run forms the baseline experiment for this paper. This baseline control run and the older POLCOMS hindcast are intercompared to evaluate how the two modeling systems perform irrespective of assimilation. They are compared with respect to satellite derived Sea Surface Temperature (SST), in situ sub surface observations as well as both global and regional climatologies. To establish the effect of the key changes from CO4 to CO5 a set of sensitivity experiments are integrated over

the full 30 year period. The key differences of the 30 year experiments are listed in Table 1.

The shorter CO4 experiments in O'Dea et al. (2012) used direct fluxes from NWP atmospheric forcing at VN3.2 of NEMO. The Met Office NWP forcing dataset only covers November 2006-2012. Thus, to investigate the effect of the different surface forcing a second set of experiments was integrated. The key differences are shown in Table 2. This second set of experiments also determines the difference between upgrading the NEMO code, and keeping all other parameters as similar as is feasible.

In the CO4 experiments there was also a bug involving the application of the inverse barometer at the lateral boundaries and its effect is explored in the 5 year experiments by re-inclusion in one VN3.4 experiment.

### 4.1   Model Initialisation and forcing

CO5 was initialised in January 1981 by interpolating temperature and salinity fields from the 1/4° ORCA025 hindcast of the standard global ocean configuration GO5.0 (Megann et al., 2014). GO5.0 was itself initialized from a mean of the EN3

monthly objective analysis (Ingleby and Huddleston, 2007) and integrated from 1976 to 2005. The lateral open ocean boundary conditions for 1981 through to 1989 were also taken from the GO5.0 hindcast. However, the boundary conditions from 1989 onwards were taken from the Global Seasonal Forecast system version 5 (GLOSEA5) (MacLachlan et al., 2015). GLOSEA5





was chosen for this period as it includes data assimilation. Unfortunately, there was no continuous run of GLOSEA5 that covered all of 1989-2012. Instead there were only two separate runs of GLOSEA5 available. The first GLOSEA5 run covered 1989-2003 and the second covered 2003-2012 . The different global models all had different mean Sea Surface Height (SSH) which needed to be matched as close as feasible to limit jumps at the cross over dates. Furthermore, both the GO5.0 hindcast

and the first four years of the GLOSEA5 integration have substantial drifts that needed to be removed. Details on the drift removal are given in Appendix B. From 1993 onwards GLOSEA5 is constrained by assimilation of altimeter data and no SSH drift removal is required over this period. NSBS GETM data at 1 nautical mile resolution was made available from IoW for the years 2000-2012. For years prior to this an annual climatology was created based on the 2000-2012 NSBS GETM data. In the control run river forcing from E-HYPE data is utilized for the full 30 year hindcast. The sensitivity experiments include

hindcasts with the climatological rivers and climatological Baltic boundary to understand the impacts of the newer inputs.

## 5 Results

### 5.1 Tidal Harmonics

The co-tidal chart of the M2 SSH tidal harmonic as analyzed from CO4 and CO5 is given in Fig. 4. Overall the general representation is fairly similar. However, the degenerate amphidrome in southern Norway is closer to the coast and better

represented in CO5. It is found that the change in the bathymetry and land sea mask due to the new Baltic boundary condition is the main driver behind the shift in the amphidrome.

Harmonic analysis of CO5 surface elevation is compared against tide gauge and bottom pressure data from the British Oceanographic Data Centre (BODC). RMS errors of model SSH amplitude and phase is shown in tables 3 and 4. The CO5 configuration as used in all sensitivity experiments in this paper has a slightly larger RMS error in both amplitude and phase

compared to CO4. Two issues behind this increase in error were found. One was due to an order of calculation bug in the time splitting in CO5. This resulted in a small error in the surface pressure gradient term. The second was in relation to the reference density within NEMO. In the CO4 configuration the reference density was $1027 \ \mathrm{kg \ m^{-3}}$. However, in CO5 the NEMO VN3.4 default of $1035 \ \mathrm{kg \ m^{-3}}$ was used. When these were corrected for, CO5 slightly improves upon CO4 when compared to the standard observations. To understand if these changes have any significant impact on the control and reanalysis a further

experiment with the changes was integrated. No significant difference in mean temperature or salinity fields was found.

### 5.2 Surface Biases

#### 5.2.1 Seasonal SST biases

The mean seasonal model SST from 1985-2004 is compared with remotely sensed products. These include the National Oceanic and Atmospheric Administration (NOAA) Advanced Very High Resolution Radiometer (AVHRR) product (Casey

et al., 2010) and the European Space Agency (ESA) Climate Change Initiative (CCI) product (Merchant et al., 2014). The period 1985-2004 is chosen for two reasons. First it allows for the CO5 hindcast to be spun up from rest in 1981. Secondly



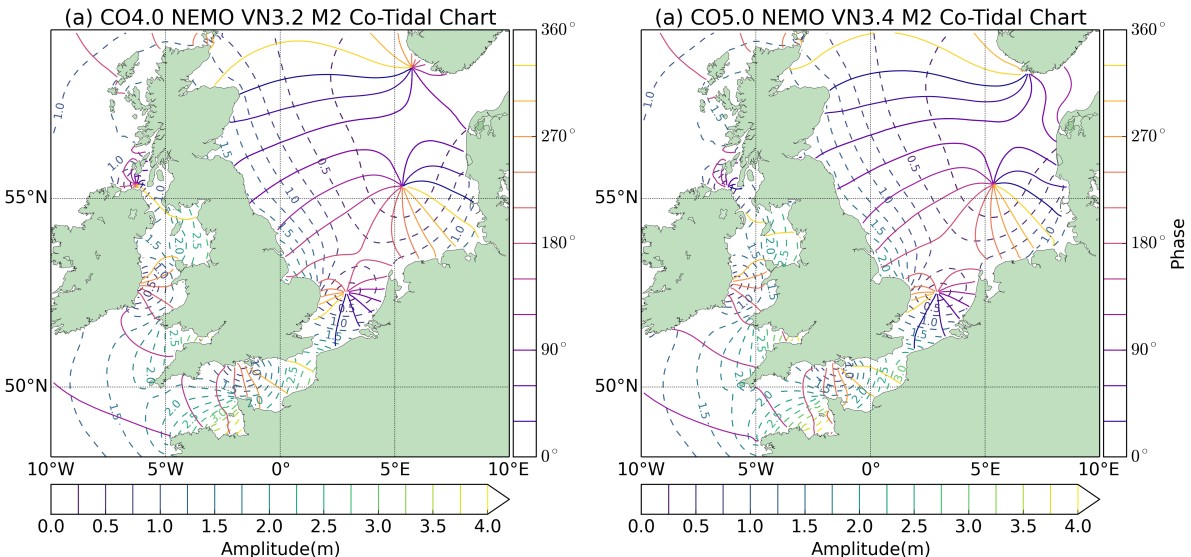

**Figure 4.** M2 co-tidal charts. (a) CO4, NEMO VN3.2. (b) CO5, NEMO VN3.4.

it presents a common period with which to compare the POLCOMS hindcast that ends in 2005. Figure 5 compares the CO5 control and POLCOMS hindcast SST bias against the AVHRR data. The largest bias in CO5 SST is the cold bias extending from eastern Iceland southeastwards to the Faroe-Shetland Channel (FSC) and from the FSC northwestwards to the northern boundary of the domain. This SST bias is less apparent in summer as seen in Fig. 5(c). The reduction in the bias might be

caused by over stratification in summer. The regions immediately surrounding the cold bias area appear to be warm biased in summer. This suggests the cold bias may be of a remote origin such as the boundary condition. Elsewhere off shelf there is a smaller cold bias in winter, spring and autumn. Along the Celtic shelf break there is a slight warm bias. The model is probably underestimating the cold water surface signal associated with enhanced vertical mixing at the shelf break. In summer off shelf southward of 50° N CO5 appears to be too warm. On shelf CO5 SST is slightly cold biased in most regions for most

seasons. However, there are some warm biases, particularly in summer. The Southern Bight, the Western Isles of Scotland and the western Irish Sea all have summer time warm biases. The English Channel also has a warm SST bias in autumn.

    Fig. 5(e)-(h) show the equivalent seasonal SST bias for the POLCOMS hindcast. POLCOMS also has a large cold bias from Iceland to the FSC and from the FSC to the northern boundary in winter, spring and to some extent in autumn. However, the POLCOMS SST cold bias appears to be more extensive. It also extends southwestwards from the FSC to roughly the Porcupine

Bank. Near the western boundary there is also a significant warm SST bias in POLCOMS north of 55° N in winter. Off shelf in summer, there is a large warm bias in POLCOMS across much of the domain. However, there is also a large summertime SST cold bias in the Norwegian Trench, the Skagerrak and the Kattegat.





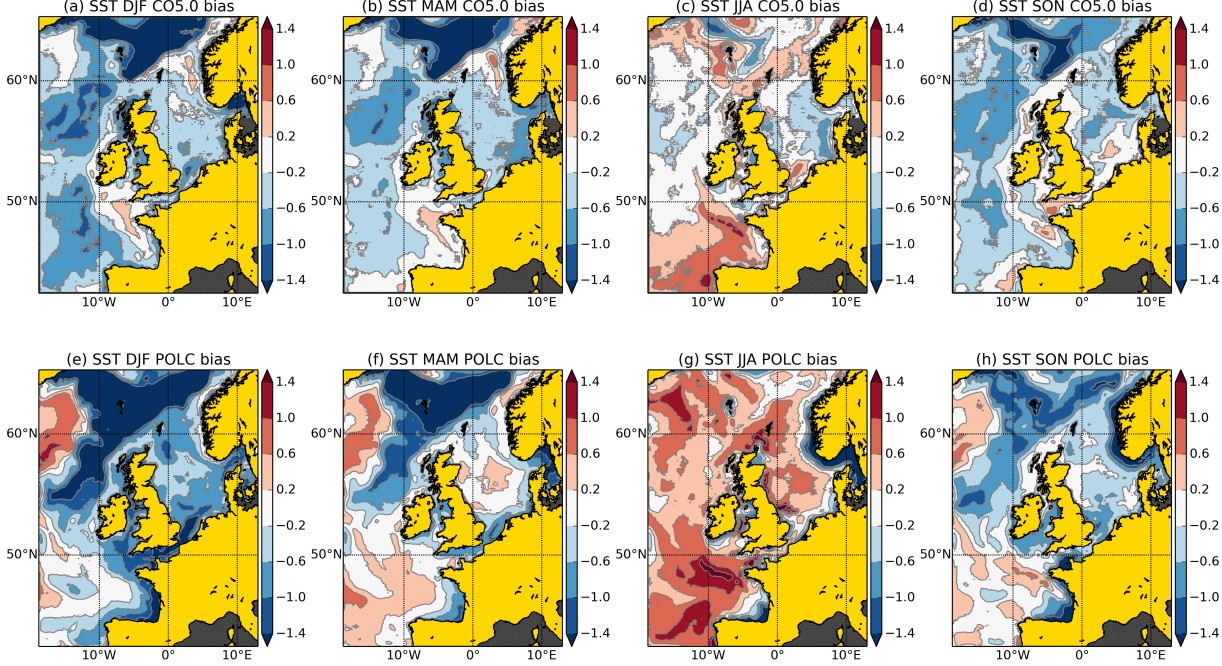

**Figure 5.** The difference between the mean seasonal model SST and the mean satellite SST for 1985-2004. (a) CO5 December-January-February (DJF) bias, (b) CO5 spring March-April-Mam (MAM) bias, (c) CO5 summer June-July-August (JJA) bias, (d) CO5 autumn bias September-October-November (SON), (e) POLCOMS (POLC) winter (DJF) bias, (f) POLC spring (MAM) bias, (g) POLC summer (JJA) bias, (h) POLC autumn bias (SON).

In summary, the CO5 control hindcast appears to have a much smaller SST bias than the preceding POLCOMS hindcast. One particularly large bias in CO5 is the large cold bias in the northern part of the domain which is also present in POLCOMS. This bias is explored further with comparisons against temperature and salinity profiles, as well as climatology. CO5 does appear to be too warm off shelf in summer but much less so than POLCOMS. On shelf CO5 is generally slightly cold biased,
5   whereas POLCOMS alternates from a large wintertime cold bias to a large summertime warm bias. POLCOMS is too cold in the Norwegian Trench during summer, while CO5 appears to do reasonably well here.

### 5.2.2 Surface Salinity biases

The mean Sea Surface Salinity (SSS) of CO5 for 1985-2004 and the POLCOMS hindcast are compared against the World Ocean Atlas 2013 (WOA13) global climatology (Zweng et al., 2013), the KLIWAS North Sea Climatology (KLIWAS) (Bersch
10   et al., 2013) and EN4 (Good et al., 2013) profile data in Fig. 6. A similar pattern in negative SSS bias as SST bias from Iceland to the FSC and to the northern boundary is present in CO5. With the exception of this northern region, CO5 off shelf is in





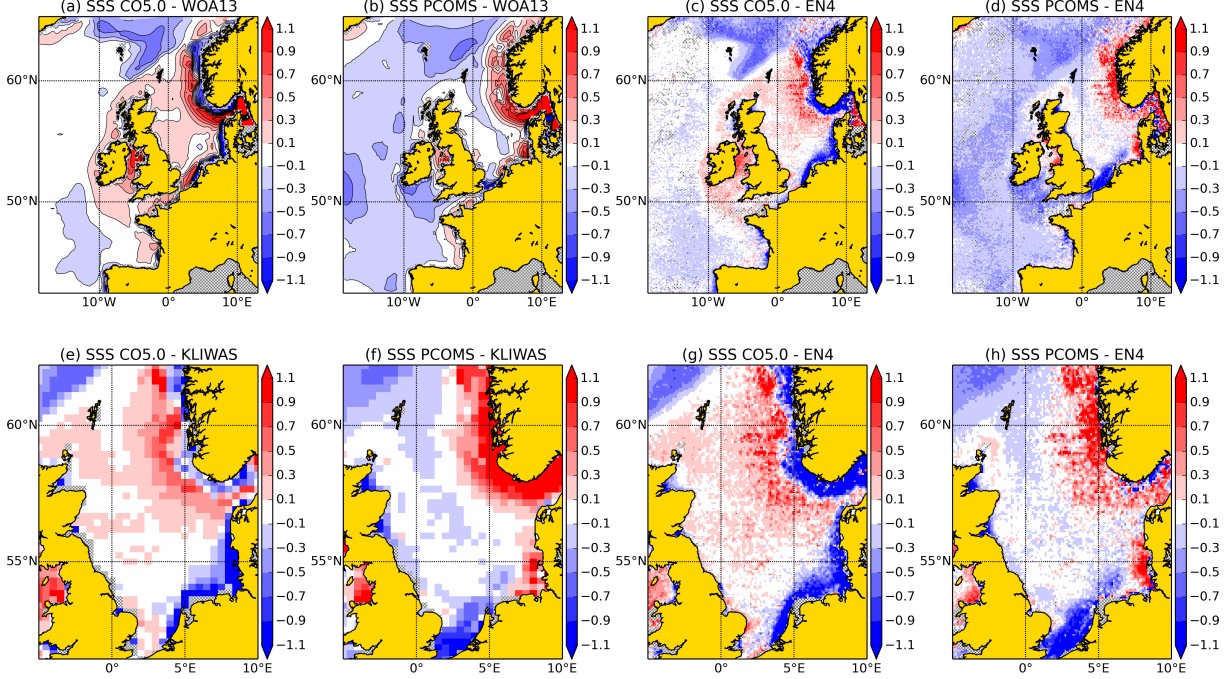

**Figure 6.** Mean model Sea Surface Salinity (SSS) differences 1985-2004 from WOA13, KLIWAS and EN4. (a) CO5 - WOA13 Climatology. (b) POLCOMS - WOA13 Climatology. (c) CO5 - EN4 (d) POLCOMS - EN4. (e) CO5 - KLIWAS North Sea Climatology. (f) POLCOMS - KLIWAS North Sea Climatology. (g) CO5 - EN4 in the North Sea. (h) POLCOMS - EN4 the North Sea.

reasonably good agreement with both the climatology and the mean profiles. However, POLCOMS appears too fresh off shelf except along the western French coast indicating an offset in surface salinity between CO5 and POLCOMS. On the shelf CO5 is in general slightly too saline. In particular, the Irish Sea is saline biased in CO5 and indicates the E-HYPE freshwater flux may be too small here.

5    Both CO5 and POLCOMS have large SSS biases compared to the climatologies and profiles in the Norwegian Trench. POLCOMS is too saline in the Norwegian Trench, while the salinity bias in CO5 is a dipole: near the Norwegian coast it is too fresh and near the western limb of the Norwegian Trench it is too saline. In POLCOMS not only are there fewer vertical levels but the vertical resolution near the surface is proportional to the ocean depth as in CO4. Consequently, compared to CO5 the surface resolution in POLCOMS in the Norwegian Trench is much reduced. The surface resolution in POLCOMS 10   over the Norwegian Trench is typically 4 to 5 metres compared to the uniform 1 meter resolution for CO5. This may account for the much more saline SSS in POLCOMS here. The Baltic boundary in POLCOMS is also more similar to CO4 than CO5 using climatological river points to represent the exchange with the Baltic. The sensitivity experiments below investigate the effect of both these changes within the NEMO framework. With regards to the dipole in CO5, the resolution at 7 km is not





sufficient to resolve the intense mixing processes in the trench where northward flowing fresh water of Baltic origin along the Norwegian coast mixes laterally with adjacent incoming southward flowing saline Atlantic water. It is anticipated that with increased horizontal resolution better representation of eddy-induced mixing may reduce the dipole here.

POLCOMS and CO5 have biases of opposite signs in the German Bight, CO5 is too fresh and POLCOMS is too saline.
POLCOMS uses the climatological rivers as in CO4 in contrast to the E-HYPE rivers used in CO5. Thus, the sensitivity experiments S30_1, S30_2 and S30_3 that compare the different river sources should help to understand the difference in this bias. POLCOMS also appears to be too fresh in the Southern Bight and this may be contributing to the saline bias in the German Bight. POLCOMS may not be advecting the Rhine outflow to the east close enough to the coast. CO5 in contrast appears to be too fresh in the vicinity of the Rhine outflow.

**5.2.3   Off Shelf Temperature and salinity biases through depth against WOA13 Climatology**

To assess how CO5 and POLCOMS behave throughout the water column off shelf they are compared against WOA13 data. Figure 7 displays both zonal transects and depth level temperature biases for 1985-2004 compared to WOA13. Both CO5 and POLCOMS temperature biases are included in Fig. 7. Figure 8 is the equivalent salinity plot. As the mean is for the entire period seasonal biases such as in the SST plots of Fig. 5 are not discernible. The location of the transects are chosen to intersect
regions of particularly large bias. Note that these comparisons use the CMEMS POLCOMS dataset, which was interpolated onto standard depth levels from the native POLCOMS grid which uses 40 s-levels in the vertical (Holt et al., 2012). The interpolated POLCOMS data is particularly coarse at depth which is reflected in the step like nature of the POLCOMS bias plots at depth.

In Fig. 7 the first two columns are zonal transects of difference in the mean temperature from the WOA13 climatology
over the period 1985-2005. The first column is for CO5 and the second POLCOMS. The geographical extent of the biases highlighted in the transects are shown for 4 depths in the last two columns of Fig. 7. Both CO5 and POLCOMS have a cold water bias centered around roughly 1000 m that originates near the southern boundary away from the relaxation zone. A warm temperature bias surrounds the cold temperature bias away from the coast. A similar pattern in salinity bias is shown in Fig. 8. It appears the models are diffusing both horizontally and vertically the warm and saline waters of Mediterranean origin entering
the domain from the southern boundary. The extra diffusion in the relaxation zone and the relatively coarse vertical resolution of about 100 m at a depth of 1000 m may be contributing to the loss in identity of the Mediterranean waters. The anomaly is also present in the Bay of Biscay but is much reduced in CO5 further north.

In the seasonal SST anomalies a large cold bias was shown in both CO5 and POLCOMS in winter. This cold bias is also present with respect to the WOA13 climatology. In CO5 and POLCOMS it extends down to around 500 m. There is a warm
bias in CO5 along the sea bed of the Iceland-Faroe ridge at around 500 m, and at a similar depth on the Shetland side of the FSC. The vertical resolution of POLCOMS is quite coarse at this depth. However, it suggests that at depths greater than 500 m POLCOMS is warm biased in the FSC and Norwegian Sea, CO5 appears to be close to the climatology below 500 m. There is a similar pattern in the salinity bias with both CO5 and POLCOMS relatively fresh near surface in this region. However,





**Figure 7.** CO5 and POLCOMS temperature bias compared to WOA13 1985-2005. Panels (a), (e), (i) and (m) are CO5-WOA13 temperature bias transects along 42° N, 45° N, 58° N, and 63° N. Panels (b), (f), (j) and (n) are POLCOMS-WOA13 temperature bias transects along 42° N, 45° N, 58° N, and 63° N. Panels (c), (g), (k) and (o) are the CO5-WOA13 temperature bias at depths 0 m, 100 m, 1000 m and 2000 m. Panels (d), (h), (l) and (p) are the POLCOMS-WOA13 temperature bias at depths 0 m, 100 m, 1000 m and 2000 m.





POLCOMS appears to be slightly fresher than WOA13 off shelf right through depth for most of the domain. Off shelf away from Biscay and the northern boundary CO5 salinity is quite similar to WOA13.

### 5.2.4   North Sea Temperature and salinity biases through depth

The KLIWAS climatology for the NWS in combination with EN4 provides an alternative to WOA13 for evaluation of the
models on the shelf itself. Figures 9 and 10 compare CO5 with both the KLIWAS climatology and the EN4 data over the period 1985-2004. A comparison of POLCOMS against KLIWAS is also included as a reference. Figure 9 focuses on the summer months when there is seasonal thermal stratification, while Fig. 10 is the salinity mean for all seasons. Including all seasons allows for a larger number of in situ profiles to compare against. In addition to biases at depth levels 10 m, 30 m and 40 m, transects are taken through areas of significant bias to give an overview of the vertical structure in the model bias.

Generally the structure of the temperature bias between CO5 and EN4 is in reasonable agreement with the structure of the bias between CO5 and the KLIWAS climatology. In the seasonally stratified areas of the North Sea, CO5 compares favorably near surface compared to POLCOMS. POLCOMS here is significantly warm biased. Immediately below the thermocline both CO5 and POLCOMS are cold biased with the cold bias in POLCOMS being somewhat larger than CO5. In CO5 the cold bias does not extend to the bed and in fact reverses sign to be warm biased near bed, whilst in POLCOMS the cold bias reduces
towards the bed with only a small bias remaining at the sea floor. The light attenuation scheme in CO5 and POLCOMS are quite different and may partially explain why POLCOMS is more warm biased at the surface and more cold biased at depth. The light scheme used in POLCOMS (PDWL) is also implemented in CO4 and is included in the sensitivity experiments to enable its impact to be assessed.

The CO5 salinity bias against EN4 is also broadly in agreement with the bias against the KLIWAS climatology. As in the
surface plots of Fig. 6, over most of the North Sea CO5 is slightly too saline through depth. Along the coasts of Holland, Germany and Denmark CO5 is clearly too fresh, suggesting too much riverine input as discussed earlier. Away from the coasts, POLCOMS is in fairly good agreement with EN4 and KLIWAS while just slightly fresher at depth. The transects in Fig. 10 are taken to go through the Norwegian Trench and the Rhine plume. CO5 is shown to be roughly 0.5 too fresh above 20 m in the Norwegian Trench near the coast of Norway, while below 20 m CO5 is slightly too saline. The warmer and more saline water
from the Atlantic appears to make CO5 too saline along the rim of the Norwegian Trench. In contrast, POLCOMS is shown to be typically greater than 1.1 too saline above 20 m in the Norwegian Trench, while below 40 m POLCOMS switches from the large saline bias to a significantly fresh bias. It appears that CO5 is representing the haline stratification in the Norwegian Trench with greater fidelity than POLCOMS. Both the vertical resolution and the Baltic boundary condition may play some role in this and are included in the sensitivity experiments that follow.



**Figure 8.** CO5 and POLCOMS salinity bias compared to WOA13 1985-2005. Panels (a), (e), (i) and (m) are CO5-WOA13 salinity bias transects along 42° N, 45° N, 58° N, and 63° N. Panels (b), (f), (j) and (n) are POLCOMS-WOA13 salinity bias transects along 42° N, 45° N, 58° N, and 63° N. Panels (c), (g), (k) and (o) are the CO5-WOA13 salinity bias at depths 0 m, 100 m, 1000 m and 2000 m. Panels (d), (h), (l) and (p) are the POLCOMS-WOA13 salinity bias at depths 0 m, 100 m, 1000 m and 2000 m.





**Figure 9.** North Sea temperature bias compared to EN4 and the KLIWAS climatology for Summer (JJA) Panels (a) - (d) compare CO5 and EN4 at 10 m, 30 m, 40 m and along a transect at 56 ° N. Panels (e) - (h) compare CO5 and KLIWAS climatology at 10 m, 30 m, 40 m and along a transect at 56 ° N. Panels (e) - (h) compare POLCOMS and KLIWAS climatology at 10 m, 30 m, 40 m and along a transect at 56 ° N. Panels (m) - (o) are transects through depth for each case along longitude 2.8 ° E.







**Figure 10.** North Sea salinity bias compared to annual EN4 and KLIWAS climatology . Panels (a) - (d) compare CO5 and EN4 at 6 m, 20 m, 30 m and along a transect at 58° N. Panels (e) - (h) compare CO5 and KLIWAS climatology at 6 m, 20 m, 30 m and along a transect at 58° N. Panels (e) - (h) compare POLCOMS and KLIWAS climatology at 6 m, 20 m, 30 m and along a transect at 58° N. Panels (m) - (o) are transects through depth for each case along longitude 4.8° E.

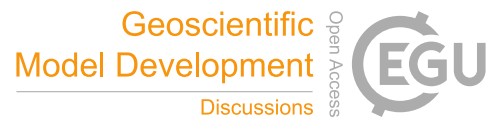


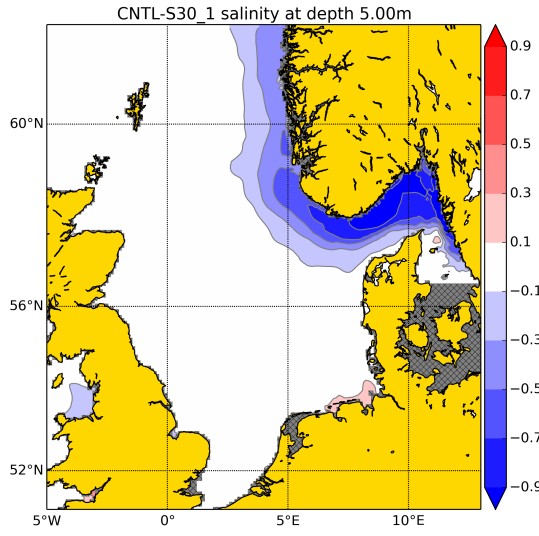

**Figure 11.** Comparison of mean salinity at 5 m between CO5 with 51 vertical levels using the Siddorn and Furner stretching function (CNTL) and 33 vertical levels using the Song and Haidvogel stretching function (S30_1).

## 5.3 30 year sensitivity runs

### 5.3.1 Vertical levels and stretching function

The effect of the changes of surface vertical resolution between CO4 and CO5 is shown in Fig. 2. Sensitivity experiment S30_1 is exactly the same as the control (CNTL) except it uses 33 SH vertical levels instead of 51 SF levels. Although there are some

5 small changes in summer time stratification off shelf, the most dramatic change concerns the surface salinity in the Norwegian Trench. Figure 11 is the difference in salinity at 5 m between the control experiment CNTL (SF51) and sensitivity experiment S30_1 (SH33). The extra vertical resolution in the control run results in less diffusion of the surface fresh layer with depth. The POLCOMS hindcast also has much less vertical resolution at the surface than CO5 and may be one factor underlying its saline bias in the surface waters of the Norwegian Trench.

### 5.3.2 Baltic and rivers

Both the river forcing and Baltic boundary condition are changed in CO5 from climatological inputs to E-HYPE riverine inputs and IoW Baltic boundary data. The 30 year sensitivity experiments S30_2 and S30_1 are intercompared in Fig. 12. S30_1 is a 33 level version of the CO5 control but with exactly the same E-HYPE rivers and IoW Baltic boundary. S30_2 is exactly the same as S30_1 except that it uses the older climatological inputs for rivers and Baltic boundary as used in CO4. Figure

12(a) shows the surface salinity bias against EN4 data for S30_1. Figure 12(b) is the same but for S30_2 and shows a large





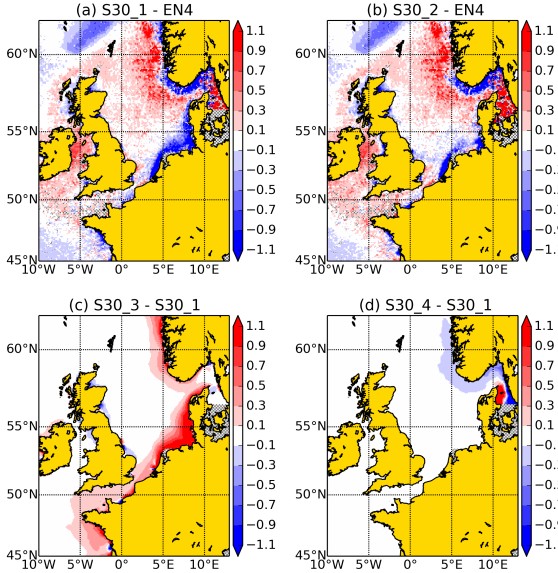

**Figure 12.** Comparing SSS using climatological river and Baltic inputs against E-HYPE rivers and IoW Baltic. Panel (a) 33 SH levels with E-HYPE rivers and IoW Baltic (S30_1) Vs EN4. Panel (b) 33 SH levels with climatological rivers and climatological Baltic (S30_2) Vs EN4. Panel (c) E-HYPE rivers minus climatological rivers (S30_4-S30_2). Panel (d) IoW Baltic - climatological Baltic (S30_4-S30_1).

reduction in the fresh water bias in the German Bight. Figure 12(c) compares experiment S30_3 with S30_1 to show differences created by the change in rivers alone. For most of continental Europe the E-HYPE rivers clearly have a greater discharge. The difference is pronounced in the German Bight contributing to the fresh bias compared to EN4 data here. Around the coast of the U.K. the E-HYPE river discharge is slightly less than the climatology. Figure 12(d) compares experiment S30_4 with

S30_1 to show the the impact resulting from the different Baltic boundaries. The IoW boundary results in a slightly more saline SSS over in the Norwegian trench. The effect of the Baltic boundary condition is much smaller than the freshening due to the E-HYPE rivers resulting in an overall freshening compared to the climatologies.

### 5.3.3 Light attenuation

The summer time biases in temperature were shown to be significantly different between POLCOMS and CO5. Sensitivity

experiment S30_5 explores replacing the light attenuation scheme in CO5 with the PDWL scheme. Figure 13 compares the control experiment and experiment S30_5 over summer. Figure 13(a) compares each run on shelf in regions of seasonal stratification. Using the PDWL light scheme has three effects; it increases the warm surface temperature bias, it increases the mid depth cold bias and it reduces the near bed bias. The partition of solar radiation into a penetrating part and a non-penetrating part is dealt with differently in each scheme and influences the degree of bias at the surface. In the PDWL scheme

all of the non-penetrating part is added to the surface layer, while in the RGB scheme there is still a slight penetration of the





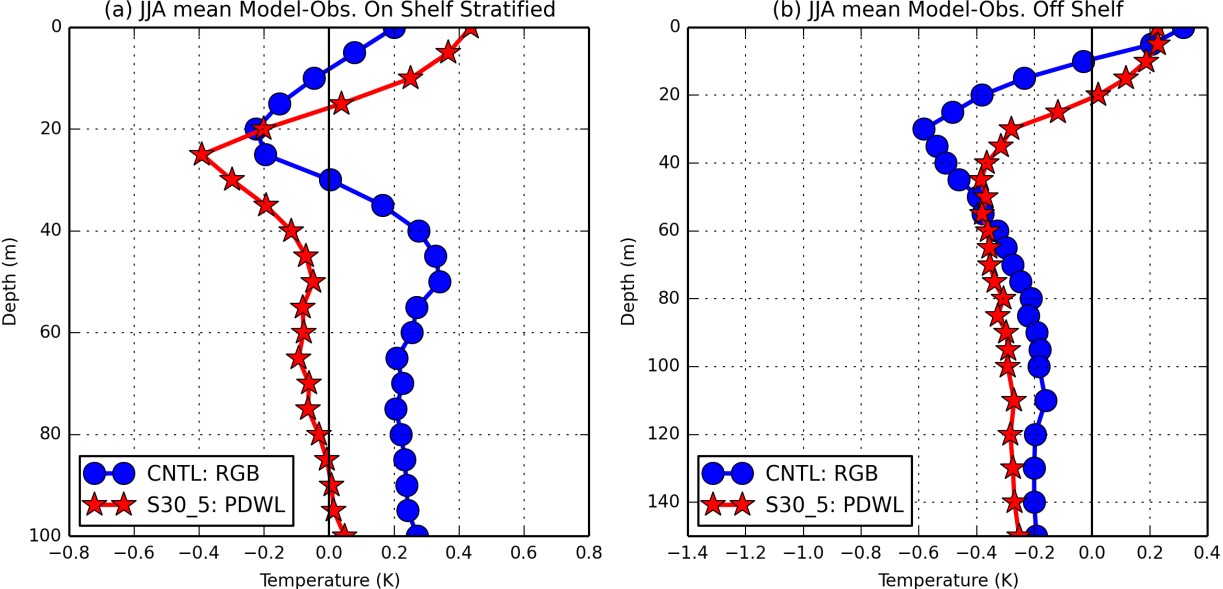

**Figure 13.** Comparison of summer time profiles compared to NEMO's three band light attenuation scheme (RGB) and POLCOMS 1 band scheme (PDWL). Panel (a) compares CNTL and S30_5 against the mean of profiles in the seasonally stratified part of the domain on shelf. Panel (b) compares CNTL and S30_5 against the mean of profiles off shelf south of 60° N.

quickly attenuating light. The cold bias in both models indicates that the depth of the thermocline is too shallow, which could be either be due to the light not penetrating far enough in both schemes or insufficient vertical mixing. At depth the PDWL scheme results in less heat being mixed down, resulting in a better agreement with the bed temperature as the RGB scheme is biased warm here. However, in mixed areas on shelf both models appear to be too warm which may indicate a bias in the

5 surface flux forcing.

Figure 13(b) compares each scheme against the mean of the profiles off shelf south of 60° N. Figure 13(b) does not show depths below 140 m as the differences due to light below this depth is negligible.The large cold bias in the upper layers of the ocean north of 60° N biases the whole field cold. Thus, to obtain a better representation of the effect of the light scheme in the absence of large underlying biases we restrict the mean to south of 60° N. As the light penetrates more deeply off shelf in the

10 PDWL light scheme, the warm bias at the surface is less than the RGB scheme and the cold bias below 20 m is also reduced. Both schemes are similarly cold biased below 60 m where the direct effect of light penetration is small.

## 5.4   5 year sensitivity runs

The shorter CO4 experiments of O'Dea et al. (2012) used different open ocean boundary conditions, and surface boundary conditions relative the CO5 control run. To further explore CO4 and CO5 differences whilst using the same forcing conditions

of CO4, a set of sensitivity experiments for 5 years were undertaken starting in November 2006. The constraint on the start date



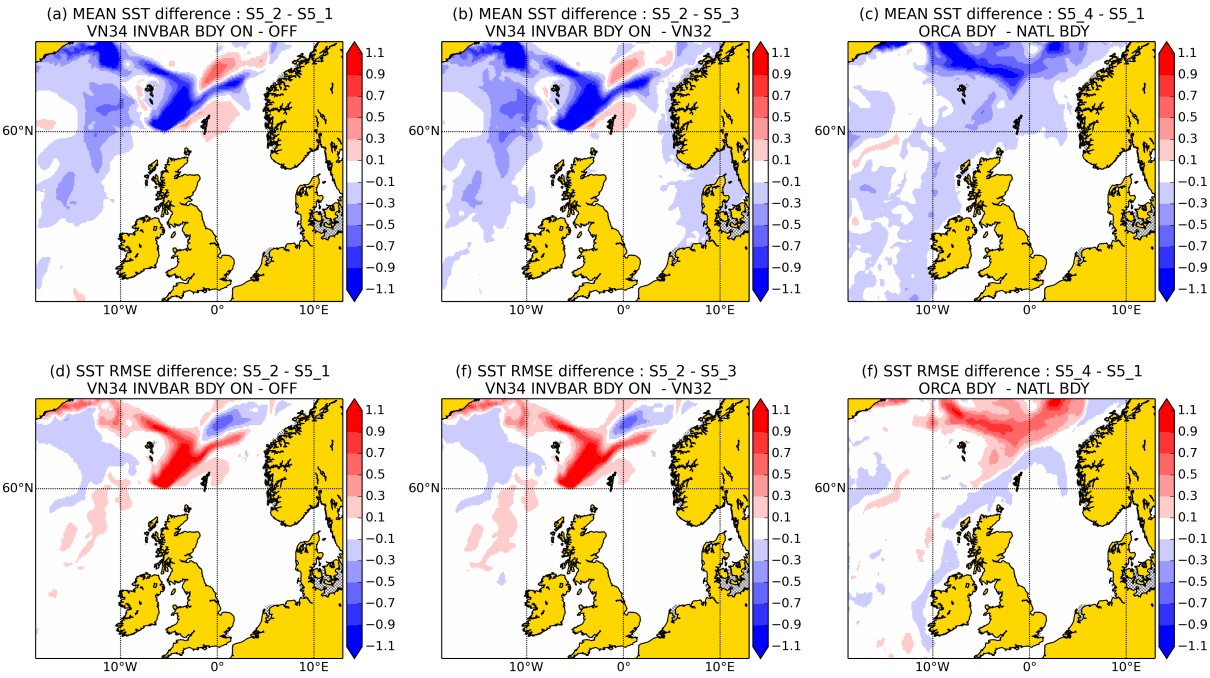

**Figure 14.** Isolating the difference in northern SST bias between CO4 and CO5. Panel (a) Mean SST difference between NEMO VN3.4 inverse barometer applied to boundary and without (S5_2-S5_1). Panel (b) Mean SST difference between NEMO VN3.4 inverse barometer applied to boundary minus NEMO VN3.2 (S5_2-S5_3). Panel (c) Mean SST difference between NEMO VN3.4 with ORCA025 and NATL12 boundary data (S5_4-S5_1). Panel (d) Difference of SST RMSE between NEMO VN3.4 inverse barometer applied to boundary and without (S5_2-S5_1). Panel (e) Difference of SST RMSE between NEMO VN3.4 inverse barometer applied to boundary and NEMO VN3.2 (S5_2-S5_3) Panel (f) Difference of SST RMSE between NEMO VN3.4 with ORCA025 and NATL12 boundary data (S5_4-S5_1).

is the availability of Met Office NWP flux data and the open open boundary conditions used in CO4 which start from November 2006. All the 5 years experiments as detailed in Table 2 have 33 vertical levels with the Song and Haidvogel stretching function (Song and Haidvogel, 1994). They also use climatological rivers, climatological Baltic, and the single band light attenuation scheme implemented in CO4. The sensitivity of the model to the vertical coordinate, rivers, Baltic boundary and the light
5   attenuation scheme is explored in the 30 year experiments. They are not shown to have significant impact on the large SST bias from Iceland to the Faroes. In the following sections the effects of changed boundaries and fluxes with an emphasis on the sensitivity of the SST bias to these changes is detailed.



### 5.4.1 Inverse Barometer and open ocean boundary condition

The 5 year sensitivity experiments show that the most significant differences between CO4 and CO5 are related to the lateral boundary conditions. A bug in NEMO VN3.2 prevented the application of the inverse barometer effect on the open ocean lateral boundaries. Thus, two sensitivity experiments with NEMO VN3.4 were conducted, S5_1 with this bug deliberately included and S5_2 without. An additional experiment, S5_3, is an equivalent experiment with exactly the same forcing but with NEMO V3.2 as the base model.

The resulting 5 year mean SST difference between S5_1 and S5_2 is shown in Fig. 14(a). Clearly the switching on or off of the inverse barometer on the open boundary has a large impact on the SST in the north of the domain. The difference between the SST RMSE of S5_1 and S5_2 shown in Fig. 14(d). The much larger RMSE of S5_1 indicates that the inclusion of the inverse barometer effect on the boundary considerably reduces the SST skill here. However, if the inverse barometer is not included on the boundaries anomalous northward flowing boundary jet currents result. Figures 14(b) and (e) are the equivalent mean and RMSE differences between S5_2 and S5_3, which are very similar to that of Fig. 14(a) and (d). The difference (not shown) between S5_1 and S5_3 is much smaller. Thus, a large component of the difference between CO4 and CO5 is the difference in the application of the inverse barometer effect on the lateral boundary.

Another significant difference was the open ocean source data interpolated onto the open boundaries of CO5 and CO4. The 30 year sensitivity experiments of CO5 used data from the 1/4° global ocean domain (ORCA025). However, CO4 was forced using a 1/12° model of the North Atlantic (NATL12). In operational implementation of CO5 the higher resolution NATL12 model is also used to derive open boundaries. Sensitivity experiment S5_4 is exactly the same as S5_1 but replaces the ORCA025 derived boundaries with boundaries derived from the NATL12 model. The mean and RMSE SST differences between S5_1 and S5_4 are shown in Fig. 14(c),(f). The NATL12 data results in a warmer SST from Iceland to the Faroes and a reduced RMSE compared to the ORCA025 data. The ORCA025 data has been shown to be anomalously cold and fresh around the Icelandic shelf compared to climatologies, suggesting the global ORCA025 model has insufficient resolution to represent the local details of the Icelandic shelf.

### 5.4.2 Surface Fluxes

Another important difference between CO4 and CO5 are the surface fluxes. In operational implementation the surface fluxes are taken from the Met Office NWP model. The experiments in (O'Dea et al., 2012) were also forced with NWP fluxes. However, comparable NWP model data is not available from before 2006 and thus the longer runs as in the CO5 control use ERAI surface fluxes. Furthermore, the NWP fluxes are directly prescribed in contrast to the CORE bulk formulae used with ERAI. A Haney correction (Haney, 1971) must also be applied when using direct fluxes with a prescribed reference SST as used by the NWP model itself.

Sensitivity experiment S5_5 is exactly as S5_1 but with ERAI fluxes instead of NWP fluxes. Figures 15(a) and (b) compare the mean SST and SSS between S5_5 and S5_1. The SST is almost uniformly warmer with NWP fluxes than ERAI fluxes, particularly in the Bay of Biscay, around the coast of the U.K. and into the Skagerrak and southern Norwegian Trench.





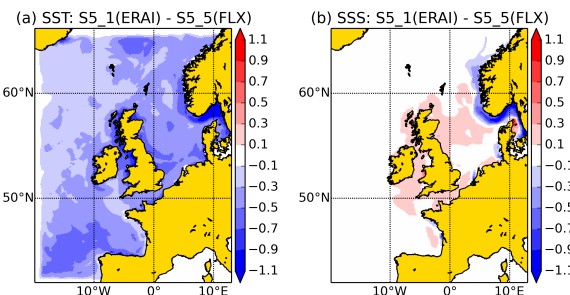

**Figure 15.** Comparison of mean SST (a) and SSS (b) differences between ERAI fluxes and NWP fluxes (S5_1 Vs S5_5).

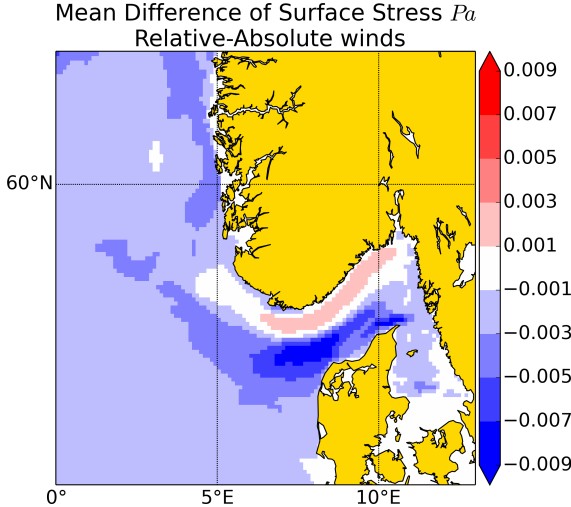

**Figure 16.** Comparison of mean surface shear stress in Pa between using wind velocity relative to ocean surface and absolute wind velocity.

However, it should also be noted that because direct fluxes use the Haney correction the resulting model SST is indirectly relaxed to the prescribed SST in a hindcast simulation. Furthermore, in NEMO VN 3.2 the surface stress from direct fluxes was based on absolute wind velocity rather than wind velocity relative to the moving ocean surface. This has important localized effects in the vicinity of persistently strong surface currents, such as the Skagerrak. This sensitivity of the model to relative winds versus absolute winds using ERAI forcing is also investigated. Figure 16 is the difference in the mean for one year of surface stress between using the relative wind velocity to the ocean surface versus the absolute wind velocity. Furthermore, the details of the fluxes near coast lines and particularly the wind stress in the Skagerrak and southern Norwegian Trench is quite





different between the lower resolution ERAI and higher resolution NWP fluxes. The differing resolution of the surface forcing and the use of absolute instead of relative wind stress is thus likely to play a role in the different sensitivity experiments.

In Fig 15(b) it is shown that the ERAI forced experiment is slightly more saline on shelf but significantly fresher in the Skagerrak and the Norwegian Trench mirroring the SST differences here. The difference in the shear stress modifies both the

transport of the surface fresh layer out of the Skagerrak and the transport of relatively saline water from the North Sea into the Skagerrak. The difference in relative and absolute winds are significant also along the shelf break from the Shetlands northwards. In this case the effect of using the absolute wind velocity is to reduce the transport of North Atlantic water northwards, which results in locally lower mean SST. With respect to the difference between ERAI and NWP forced experiments the difference in the SST in this local region is reduced. The reduction in mean difference of SST is due to the countervailing effects

of general domain wide cold bias between ERAI and NWP fluxes and the local cooling due to using absolute winds with the direct NWP fluxes.

## 6   Summary and discussion

The details of the standard coastal ocean model CO5 for the NWS were presented. CO5 was jointly developed by the Met Office and the National Oceanography Centre. This standard model forms the basis of the physics component of the current

CMEMS reanalysis of the NWS, which also includes data assimilation. CO5 is a regional tidal implementation of NEMO version 3.4, building upon upon CO4 (O'Dea et al., 2012) that used NEMO version 3.2 as the base code. In this paper a 30 year physics only control of CO5 using ERAI surface forcing and ORCA025 lateral boundary conditions has been assessed against standard climatologies and observations to understand the impact of model physics on biases. The assessment compares CO5 to a POLCOMS based hindcast over the period 1985-2004, which is a period covered by both hindcasts. A set of 30-year

sensitivity hindcasts have also been assessed to understand several changes, relative to CO4, introduced into CO5. A further set of 5-year sensitivity experiments focusing on different surface and lateral boundary conditions have also been investigated.

Overall the CO5 tides are of comparable quality to CO4. The reference density of 1035 $\text{kg m}^{-3}$ used in the control run slightly degraded the tidal predictions. The position of the degenerate amphidrome in southern Norway is slightly improved in CO5 mainly due to a slight change in the land sea mask originating from a change in the Baltic boundary condition.

Compared to AVHRR data CO5 has a large SST bias extending from Iceland to the FSC. It is particularly pronounced in winter, and partially obscured in summer due to surface heating. POLCOMS also has a large seasonal cold SST bias in the region but also a significant warm SST bias domain wide in summer. In comparison to the AVHRR observations, CO5 appears to significantly improve upon the simulation of SST in the POLCOMS hindcast.

As in the SST, CO5 has a similar pattern of fresh bias in the near surface salinity from Iceland to the FSC as well as a

large fresh bias in the German Bight and a dipole of surface salinity bias along the Norwegian Trench that suggests insufficient lateral mixing. POLCOMS in contrast is slightly too saline in the German Bight and uniformly too saline at the surface along the Norwegian Trench.



Both CO5 and POLCOMS appear to lose the identity of relatively warm saline Mediterranean water near the southern boundary of the domain. In CO5 there is a sponge layer in the boundary relaxation zone where the diffusion is increased for model stability. Furthermore, the vertical resolution is focused on the surface and bed and is particularly coarse mid-water in the deeper parts of the domain. Both of these may be contributing to the apparent overestimation of diffusion of this water mass

both laterally and vertically.

In the North Sea there is a marked difference in the vertical summer temperature profile between POLCOMS and CO5 in seasonally stratified regions. Compared to climatology and observations, POLCOMS is much too warm at the surface, while both POLCOMS and CO5 are too cold mid-water and CO5 is too warm towards the bed.

The single band light scheme (PDWL) used in POLCOMS and CO4 was seen to significantly alter the temperature profile

in seasonally stratified regions in CO5. Introducing the PDWL scheme into CO5 leads to a larger warm bias at the surface and a larger colder mid-water cold bias than the CO5 control. Near bed the PDWL light attenuation scheme resulted in closer agreement with climatology than the CO5 control run. Both models appear to be over stratifying with a very abrupt thermocline. Whilst the light attenuation scheme may be a component of this error, the vertical mixing will also be an important contributor and should be a subject of further refinement.

The sensitivity experiments also explored the significance of changing the riverine inputs and the Baltic boundary condition. The older climatological rivers greatly reduce the fresh water bias in the German Bight and also near the Norwegian coast. It appears that the version of E-HYPE used in CO5 has too much fresh water discharge from continental Europe. The Baltic boundary condition used in CO5 results in slightly more saline surface waters in the Norwegian Trench. The added variability introduced by the CO5 Baltic boundary relative to the CO4 climatological boundary can't be assessed by the long term cli-

matological means used in this paper. Further site specific studies in the Kattegat and Skagerrak are required to evaluate the variability.

The impact of the change in vertical levels has a significant impact on the mean surface salinity in the Norwegian Trench. The increase in surface resolution allows retention of the relatively fresh layer of Baltic origin more than the coarser vertical levels used in CO4.

The 5-year sensitivity experiments revealed that a bug fix in CO5 related to the application of the inverse barometer effect on the lateral boundaries, results in a colder SST from Iceland to the FSC. This is the region where CO5 has a particularly large SST cold bias and partially explains why CO5 has larger SST errors here than CO4. The inclusion of the inverse barometer effect on the boundaries results in a greater transport of water southwards from the northern boundary, and with it colder fresher water. The source data for the boundaries themselves also have a significant impact in this region. The higher resolution $1/12°$

NATL12 model results in smaller cold bias here also. It is likely that $1/4°$ ORCA025 global model lacks sufficient resolution to model the Icelandic shelf in the vicinity of the northern CO5 boundary. The increased southwards transport of water from the northern boundary condition due to the inclusion of the inverse barometer effect amplifies the cold and fresh anomaly of the ORCA025 boundary data.

The impact of changing the surface boundary conditions from ERAI and CORE bulk forcing and directly specified fluxes

from the Met Office NWP model was also investigated. The NWP fluxes as used in CO4 resulted in a warmer mean SST




further offsetting the generally cold bias in the CO5 control off shelf. However, it also led to a slight mean warm bias on shelf with the exception of the Skagerrak where the fluxes have a fairly large cold bias. The direct fluxes as applied in CO4 used the absolute wind velocity rather than the relative wind velocity compared to the moving ocean surface. The effect of using relative versus absolute wind velocities has important impacts in localized regions with persistent strong surface currents such as the

Skagerrak. The ERAI forcing is also of a relatively coarse resolution and the details of of near coastal fluxes are quiet different from the NWP fluxes. The combined difference of absolute versus relative winds and differing details in the fluxes combine to have significant impacts in local regions such as the Skagerrak.

In summary, CO5 has been shown to produce a significantly improved hindcast of the NWS compared to POLCOMS against climatologies and observations. However, there are a number of notable biases in CO5 that need addressing in future configu-

rations. Particular issues relate to freshwater inputs from rivers, surface boundary conditions as well as seasonal stratification in the North Sea.

The next standard configuration CO6 will be an incremental update for the physics based on some of the lessons learned from CO5. The relative stability of physics developments between CO5 and CO6 allow for significant updates to both data assimilation and biology components for the NWS forecast system. Physics changes will include updating the base version of

NEMO to 3.6, updating the light attenuation to use satellite observed climatology of ocean colour instead of a domain wide coefficient. The river inputs will be from an updated climatology with reduced biases compared to the E-HYPE rivers used in CO5. However, a step change in the physics will occur in CO7 when the resolution will be increased from 7 km to 1.5 km. CO7 will be of a sufficient resolution to resolve the internal Rossby Radius on the shelf. Possible improvements include capturing to first order the generation of internal tides at the shelf break, resolving mesoscale eddies and consequently enhanced mixing

in the Norwegian Trench, greatly improved bathymetry and coastline to name but a few. Furthermore, CO7 is being developed in anticipation of the longer term goals of coupling to wave, atmosphere and land systems models. The aspiration is to drive towards eventual operational coupled implementation for which CO7 will form the basis of the ocean model component.

## 7   Code availability

The model code for NEMO V3.4 is freely available from the NEMO website (www.nemo-ocean.eu). After registration the

FORTRAN code is readily available using the open source subversion software (http:/subervsion.apache.org). Additional modifications to the NEMO3.4 trunk are required for CO5.0 and are merged into the CO5 package branch. The CO5 package branch is freely available from the NEMO repository under: branches/UKMO/CO5_package_branch.

The NEMO namelist used for CO5 is publicly available at the following DOI: 10.13140/RG.2.2.17410.89286 (O'Dea, 2016b).



# 8 Data availability

The nature of the 4D data generated requires a large tape storage facility. The data that comprises the CO5 control experiment is of the order of 6TB and each 30 year sensitivity experiment is of the same order. However, the data can be made available upon contacting the authors.

## Appendix A: FPP keys used in CO5

## Appendix B: Adjusting the lateral open ocean boundary conditions

The lateral open ocean boundary conditions are derived from three separate 1/4° ORCA025 experiments. 1981 through to 1989 are also taken from the GO5.0 1/4° ORCA025 hindcast (Megann et al., 2014). The boundary conditions from 1989 onwards are taken from the two separate Global Seasonal Forecast system version 5 (GLOSEA5) (MacLachlan et al., 2015) experiments spanning 1989-2003 and 2003-2012.

Each of ORCA025 experiments had substantially different mean SSH. They needed to be matched at the cross over dates as closely as possible to prevent large shocks. The free running model GO5 experiment for the 1980's was shown to have a long term unrealistic drift in the mean SSH. This long term trend is removed from the data before deriving boundary conditions. Furthermore, the first GLOSEA run does not have altimeter assimilation until 1992 and likewise has an unrealistic drift removed from these initial years (1989-1992).

Once the data is detrended a mean SSH is calculated area wide at the cross over dates in 1989 and 2003. The 2nd GLOSEA data set is taken as the reference. The difference in the mean SSH in the earlier detrended GO5.0 experiment at the 1989 cross over data is then subtracted from the entire period 1981-1989. This in effect is a uniform shift in SSH so that at the cross over date the discrepancy is as reduced as possible. Similarly the difference in the mean between the first GLOSEA run and the second is used to match the two in 2003. However, even after this prepossessing there is still some transient adjustment in SSH particularly so at the 2003 cross over.

## Appendix C: Other inputs

The bathymetry used in CO5 is made publicly available from the following DOI: 10.13140/RG.2.2.25799.50081 (O'Dea, 2016a).

*Competing interests.* The authors declare that they have no conflict of interest

*Acknowledgements.* Funding support is gratefully acknowledged from the Ministry of Defence, the Public Weather Service, the European Community's Seventh Framework Programme FP7/2007- 2013 under grant agreement no. 283367 (MyOcean2) and from the Copernicus



Marine Environment Monitoring Service. We acknowledge use of the MONSooN system, a collaborative facility supplied under the Joint Weather and Climate Research Programme, a strategic partnership between the Met Office and the Natural Environment Research Council. We also acknowledge the Centre for Environmental Data Analysis (CEDA) for the use of JASMIN (Lawrence et al., 2013) for post processing the model data.



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





**Table 1.** 30 Year Sensitivity Experiments

| EXP | Levels | River | Baltic | Light |
|------|--------|-------------|-------------|------|
| CNTL | SF51 | E-HYPE | IoW | RGB |
| S30_1 | SH33 | E-HYPE | IoW | RGB |
| S30_2 | SH33 | Climatology | Climatology | RGB |
| S30_3 | SH33 | Climatology | IoW | RGB |
| S30_4 | SH33 | E-HYPE | Climatology | RGB |
| S30_5 | SF51 | E-HYPE | IoW | PDWL |

CNTL is the CO5 control. SF51 refers 51 SF-levels. SH33 refers to 33 SH levels. IoW refers to data from the IoW NSBS GETM model of the Baltic (Gräwe et al., 2015). RGB is the default tri-band light attenuation scheme in NEMO Lengaigne et al. (2007). PDWL refers is the one band scheme that varies attenuation in proportion to sea bed depth (Holt and James, 2001).



**Table 2.** 5 Year Sensitivity Experiments

| EXP | NEMO | Invbar | Bdy Data | SBC |
|------|------|--------|----------|------|
| S5_1 | VN3.4 | No | ORCA025 | NWP |
| S5_2 | VN3.4 | Yes | ORCA025 | NWP |
| S5_3 | VN3.2 | No | ORCA025 | NWP |
| S5_4 | VN3.4 | No | NATL12 | NWP |
| S5_5 | VN3.4 | No | ORCA025 | ERAI |

All 5 year experiments use the single band light attenuation of Holt
and James (2001). S5_1-S5_4 use directly specified fluxes from
the Met Office Numerical Weather Prediction (NWP) model. S5_5
uses ERAI fluxes as in the CO5 control. VN3.2 and VN3.4 refer to
the base version of NEMO. Invbar specifies whether the inverse
barometer effect is is added at the boundary or not. ORCA025 and
NATL12 refer to the the source model data used for the open
lateral boundary conditions.





**Table 3.** Elevation RMSE of amplitude in cm as compared to observations.

|  | M2 | S2 | K1 | O1 | N2 |
|---|---|---|---|---|---|
| CO4 | 10.3 | 3.7 | 1.8 | 1.9 | 2.9 |
| CO5 | 11.4 | 4.5 | 2.0 | 1.9 | 3.4 |
| CO5* | 9.5 | 4.0 | 1.8 | 1.6 | 3.3 |

CO5* refers to CO5 with with lower reference density
and time-splitting bug fix



**Table 4.** Elevation RMSE of phase in degrees as compared to observations.

|      | M2   | S2   | K1   | O1   | N2   |
| ---- | ---- | ---- | ---- | ---- | ---- |
| CO4  | 14.7 | 12.8 | 17.1 | 15.7 | 21.6 |
| CO5  | 15.5 | 15.1 | 18.7 | 14.7 | 20.6 |
| CO5* | 12.6 | 11.8 | 15.4 | 14.8 | 19.2 |

CO5* refers to CO5 with with lower reference density and
time-splitting bug fix



**Table 5.** FPP keys used with CO5 control experiment

| | |
|---|---|
| key_tide | Activate Tidal Potential Forcing |
| key_dynspg_ts | Free surface volume with time splitting |
| key_ldfslp | Rotation of Lateral Mixing Tensor |
| key_iomput | Input output manager |
| key_vvl | Variable Volume Layer |
| key_shelf | Diagnostic Switch for output |
| key_zdfgls | Generic Length Scale Turbulence scheme |
| key_bdy | Use open lateral boundaries |
| key_amm | Dimensions for AMM domain. |
| key_levels=51 | Number of vertical Levels |