# Peer review of "The CO5 configuration of the 7km Atlantic Margin Model: Large scale biases and sensitivity to forcing, physics options and vertical resolution"

_Geoscientific Model Development, 2017_

## Referee Comment (RC1) · R. Hordoir (Referee) · 21 Feb 2017

Review of ÂńÂǎThe CO5 configuration of the 7km Atlantic Margin Model: Large scale biases and sensitivity to forcing, physics options and vertical resolutionÂǎÂż by O'Dea et al. - Robinson Hordoir

This article describes a new Nemo based configuration of the North West European Shelf, and compares it with previous similar configurations. The article is well written and nice to read, and as the developer of a similar configuration (Nemo-Nordic) I have actually learned a lot of things about biases that I had not thought about. I recommend this article for publication in GMD Nemo Special issue, but I have a few remarks on some points which I found not self explicit enough in the article. Further I also think

there are a few things that could be added in terms of results. Also if I had one critic on the article, it would be that the comparisons are mostly from one model to another, with a lot of changes like the vertical resolution, the boundary conditions etc.... so at the end it is a bit hard to know what causes what. A part that could be really nice would be to have also comparison of CO5 with itself but for example with different Baltic outflow parameterizations or different runoff sources etc... to provide a better understanding of how this affect processes.

Part 2: there is a mention of CO4 as a reference, which obliges the reader to read O'Dea et al 2012, so for example there is not explicit mention of the vertical grid of CO5. I guess it is z*-sigma coordinates but it would make the article a lot easier to read if this was mentioned explicitly, this part is described in many details in O'Dea et al 2012 so it would be nice to have a little summary, especially after what comes just after.

Especially, at lines 31&32 page 3: So just to be sure. Viscosity here concerns only momentum of course, and there are 2 types of viscosity at once, both applied on momentum. Which means, if I understand rightly is that the Laplacian viscosity is applied using a rotation to fit geopotentials, and Bi-Laplacian is applied using model levels, and both at the same time ? Is it possible to activate both Laplacian and Bi-Laplacian in Nemo natively or some code has been developed ? At first I thought it was a confusion between Laplacian which would be applied to diffusion and Bi-Laplacian applied to momentum, but obviously not. Anyway, this is very original, and I think it should be described in more details, what is the advantage of this method ? And by the way, how is tracer diffusion handled ?

Figure 3: It would be really nice to have mean values of all runoff datasets. And perhaps mean values per basins.

Part 5 on results. The only result concerning SSH is on the M2 tidal harmonics. Since sea level and sea level variability is a driver of the entire ecosystem I think this part

should show more results. Why not include Taylor diagrams ? And perhaps also a better assessment of the wind driven SSH ? My experience is that models have biases to represent SSH which greatly depend of the frequency. Further, SSH variability is the driver of barotropic currents like the cyclonic loop in the North Sea, having a right SSH representation affects therefore greatly the export of freshwater for example, and therefore the possible salinity biases, which can be explained not only by the amount of river runoff. Additionally, in places like the North Sea, SSH variability does not only create transport but also mixing and tidal straining which greatly affect freshwater dynamics. So the salinity bias explanation that comes after could also be related with a bias in barotropic dynamics, not only the runoff. A deeper analysis could include a computation of the North Sea gyre circulation, and that of the Strait of Dover.

Part 3: About the Baltic boundary approach using GETM, the approach is interesting but a bit heavy to carry. Basically one needs always to run GETM, and there is little estimation of the impact of having a realistic Baltic Sea outflow. It would really nice to see a real sensitivity experiment of the impact on salinity structure along the Swedish/Norwegian coasts. The Baltic Sea outflow is mostly a barotropic process driven by wind forcing over the Baltic: GETM is a cool model, but that is a lot of complexity for such a process.

---

## Referee Comment (RC2) · B. Hackett (Referee) · 26 Feb 2017

Review of "The CO5 configuration of the 7 km Atlantic Margin Model: Large scale biases and sensitivity to forcing, physics options and vertical resolution" by Enda O'Dea et al.

General comments:

This Model Development paper describes a new configuration of the NEMO code – called CO5 - applied to the European Northwest Shelf area, with the aim of producing an upgrade of the CMEMS multi-year model product for that region. The description focuses on updated model components with respect to the previous version – call CO4 – which is documented in O'Dea et al 2012. A 30-year hindcast simulation (no assimilation) is carried out with CO5 and the results are compared with the current CMEMS multi-year product, which is also a hindcast (no assimilation) produced using the POLCOMS code. The two hindcasts are also compared with observations and climatologies in order to show that the new one gives improved large-scale biases. Returning to the CO4-to-CO5 changes, an number of sensitivity runs are made to demonstrate the impact of the individual component changes; in these cases, bias and rmse for temperature and salinity, as well as tidal analyses, are presented.

The paper is well-written in very good English. The topic and material are very relevant for the development of operational oceanography in the NWS region. The sensitivity tests of NEMO component upgrades are in particular of interest to the coastal modeling community. The paper should definitely be published in GMD.

The paper really has two topics: the CO4-to-CO5 model upgrades and the CO5 – POLCOMS hindcast comparison. They are tied together by the use of 30-year hindcasts for the model sensitivity and the CMEMS product upgrade aspects, respectively. I think this is acceptable, but it demands considerable care in organising the material. I found it all a bit confusing on first read. My main recommendation is that the authors should put some effort into simplifying the terms used and explaining more carefully (repeating) what the purpose of each (sub-)section is. I also suggest adding a table with the relevant features of the 3 models: POLCOMS, CO4, CO5; it would make the similarities and differences clearer in the discussions in Section 5.

Specific comments:

1. The term AMM is used in the title and in the 1$^{st}$ para of the Introduction, and nowhere else. There should be an explicit connection to CO4 and CO5. Better yet, reduce the terminology to one term for the bulk of the paper.
2. Introduction p.2: Make clear the connection between AMM and CO5
3. p2.l6-8 and p2.l17-18: These two sentences say the same thing. Move the latter to the former's place?
4. p2.l14-17: This list of changes should include the atmospheric fluxes.
5. Section 3: Add sub-section numbers 3.1 – 3.4 at the appropriate places, e.g., p4.l17, p4.l28, …
6. p5.l1: Add url for E-HYPE data
7. p6.l13: Add a referral to fig. 11
8. p7.l5-…: Give a reason for why the surface bdry condition was changed.
9. p8.l14: "...amphidrome in southern Norway ... better represented" needs to be substantiated. Add a reference.
10. p8.l15-16: "It is found that..." without further explanation means "unsubstantiated claim" in my book. Was a sensitivity test done? Are the CO4 and CO5 bathymetries otherwise identical?
11. p12.l15-18: Is this meant as an explanation for why the bottom profiles in fig. 7 look different? If so, say so.

12. P18.Figure 11: Add "The figure shows CNTL minus S30_1. Grey area in Kattegat indicates interface to Baltic NSBS model data."
13. p19.l5: should be "slightly less saline" if the Figure 12 caption is correct.
14. p13.l13: to me "near bed" means right at the bottom, but Figure 13a shows that the PDWL scheme reduces bias over more than half the water column. Reformulate.
15. p25.l11: same as previous
16. p22.l21-23: Give a reference for the cold/fresh bias in ORCA025.
17. p22.l25-30: There appear to be two differences at play here: different sources (Met Office NWP and ERAI) and how they are applied (as direct fluxes or via CORE bulk formulae). "ERAI fluxes" is a bit misleading; use "ERAI-derived fluxes" or similar in stead. Is data availability the reason for doing this sensitivity test?
18. p22-24 Section 5.42: Why not show the SST/SSS biases in the two runs in Figure 15? I would imagine the discussion would be easier.
19. p23.l1-p24.l2 and p26.l2-7: This is interesting and could be substantiated more. What is the actual impact on the circulation? "Significant impacts" is not very useful.
20. P24.l22-24: Needs some explaining, as mentioned above.
21. P24.l29-31: CO5's SSS bias in the German Bight is also affected by E-HYPE.

Technical comments:

1. "data" is a plural and should be treated grammatically as such. Many occurrences of this error.
2. "intercompare". (A pet peeve, admittedly) Defined to mean to "compare each member of a group against all other members" (Collins Dictionary). In this paper "compare" is more appropriate, since there is only comparison of 2 members. BTW: "intercompare" is not found in the online Oxford Dictionary.
3. There are a surprising number of "is-are" errors that a technical editor should pick up.
4. p1.l4: "seamless"
5. p12.l34: Replace "However" with "On the other hand"
6. p18.l8: "CO5 and may be" → "CO5, which may be"
7. p21.l5: "...the 30 year experiments (Section 5.3)"

---

## Author Response (AR1)

Part 1 : Response to reviewers comments.
Note these are already in the discussion section but repeated here for clarity
References to Attached figures are now in the revised manuscript

**Review 1:**
1) "A part that could be really nice would be to have also comparison of CO5 with itself but for example with different Baltic outflow parameterizations or different runoff sources etc. . . to provide a better understanding of how this affect processes"

Reply:
The purpose of the article was to assess the impacts of specific changes made from the older CO4 configuration to the CO5 configuration. It first compares the configurations against climatologies and observations to present the large scale biases. It then goes on to examine the impacts of the changes bearing in mind the already presented background bias of the entire system.

With regards to Baltic outflow, there are two boundary conditions considered, that of CO4 which specifies the Baltic exchange using to River points, and in CO5 which takes temperature and salinity data from the IoW GETM model.

The impacts upon salinity to these differing Baltic boundaries are shown in Figure 12, panel (d). And the article details the impact as "The IoWboundary results in a slightly more saline SSS over in the Norwegian trench. The effect of the Baltic boundary condition is much smaller than the freshening due to the E-HYPE rivers resulting in an overall freshening compared to the climatologies."

This may be cross referenced against Figure 6 panels (e) and (g) which show the CO5 is too fresh at the surface in the Norwegian trench.

2.1)
"there is a mention of CO4 as a reference, which obliges the reader to read O'Dea et al 2012, so for example there is not explicit mention of the vertical grid of CO5. I guess it is z*-sigma coordinates but it would make the article a lot easier to read if this was mentioned explicitly, this part is described in many details in O'Dea et al 2012 so it would be nice to have a little summary, especially after what comes just after"

Reply:
This is correct, the coordinate system has not changed between CO4 and CO5. The main difference is the stretching function and the number of levels used. The details on the stretching function or within the reference "Siddorn, J. and Furner, R.: An analytical stretching function that combines the best attributes of geopotential and terrain-following vertical coordinates, Ocean Modelling, 66, 1–13, 2013." However, we explicitly state this as suggested by the reviewer to aid the flow of the manuscript.

2.2) Which means, if I understand rightly is that the Laplacian viscosity is applied using a rotation to fit geopotentials, and Bi-Laplacian is applied using model levels, and both at the same time ? Is it possible to activate both Laplacian and Bi-Laplacian in Nemo natively or some code has been developed ?

Reply:
 That is correct both forms as in CO5 are used simultaneously. The NEMO code at 3.6 does not support this natively. The relevant code changes may be found here:
ldfslp.F90
http://preview.tinyurl.com/nx7x4hc
ldfdyn_oce.F90:
http://preview.tinyurl.com/khkea3b
dynldf.F90:
http://preview.tinyurl.com/kva8bkr
dynldf_bilapg.F90:
http://preview.tinyurl.com/n7cojz
dynldf_iso:

http://preview.tinyurl.com/lllr49o
The purpose of the bilaplacian is to stabilize grid noise in the numerical solution, the purpose of the laplacian is to parameterise unresolved processes at the coarse 7km resolution.

2.3) And by the way, how is tracer diffusion handled ?

Reply:

Tracer diffusion as in CO4 is simply laplacian diffusion on geopotential levels.
Details may be found at:
https://www.researchgate.net/publication/311107924_NEMO_namelist_for_CO5_AMM7

2.4) Figure 3: It would be really nice to have mean values of all runoff datasets. And perhaps mean values per basins.

Reply:
Attached is a figure for inclusion into the updated article with the difference in the mean flow along coastal sections between the EHYPE data set used in CO5 and the climatological rivers used in CO4. It highlights the larger input of fresh water with the EHYPE dataset. This should be compared with Figure 12 Panel C that compares the model SSS using the different river sources, and again referenced back to the fresh bias of CO5 in Figure 6 panels (e) and (g). There is a strong correlation between the increased freshwater input using EHYPE and the SS fresh water bias in CO5.
Additionally the units are corrected on Fig 3

2.5) Additionally, in places like the North Sea, SSH variability does not only create transport but also mixing and tidal straining which greatly affect freshwater dynamics. So the salinity bias explanation that comes after could also be related with a bias in barotropic dynamics, not only the runoff. A deeper analysis could include a computation of the North Sea gyre circulation, and that of the Strait of Dover.

Reply:

We have conducted a computation of the transports around the North Sea using the standard NOOS transects. A plot is included in the amended article which includes the mean EHYPE river inputs within the area bounded by the transects. The transects 1, 2, 3, and 13 are in the same ball park as the observational references : NOOS 1 : Otto et al 1990: 0.6 Sv NOOS 2: Otto et al Inflow (0.7-1.11Sv) Outflow 1.8Sv NOOS 3: Otto et al 0.3 Sv NOOS 13: Prandle et al 1996 0.094 Sv

The Net flow at NOOS Transects 9 and 2 will be incorrect ( too low) as the barotropic boundaries are not specified in CO5 for the Baltic.
The transports of the North Sea circulation look reasonable which adds weight to the riverine inputs being the leading order term in the salinity bias for CO5. The revised paper includes this first order analysis of the transports, but a more detailed analysis the transport should be the subject of a separate paper.

3 ) About the Baltic boundary approach using GETM, the approach is interesting but a bit heavy to carry. Basically one needs always to run GETM, and there is little estimation of the impact of having a realistic Baltic Sea outflow. It would really nice to see a real sensitivity experiment of the impact on salinity structure along the Swedish/Norwegian coasts. The Baltic Sea outflow is mostly a barotropic process driven by wind forcing over the Baltic:

Reply:

It would definitely be of interest to compare the effect of the various Baltic model data in the Skagerrak and Kattegat.
However, the barotropic processes in CO5 will not be correctly represented in the Skagerrak/ Kattegat due to the simplistic boundary conditions used. The model is likely to do poorly here and doesn't warrant further investigation. The GETM model is not used for operational forecasts but at the model data was readily available from hindcasts it was used for the reanalysis. Baltic CMEMS data is used operationally. Future higher resolution configurations (1.5km CO7) will move the boundary south of the Danish Straits and used full boundaries. It will certainly be of value perform a detailed analysis of this system with a variety of source boundary data.

Review 2:

1) The term AMM is used in the title and in the 1st para of the Introduction, and nowhere else. There should be an explicit connection to CO4 and CO5. Better yet, reduce the terminology to one term for the bulk of the paper

Reply: CO5 is in effect a version of the AMM. AMM is the modelling system for the of the NWS run at the Met office and dates back to POLCOMS models of the same region. CO5 is a particular version of AMM, and CO4 was it's immediate predecessor with which we compare against. The previous version of a long hindcast was only done with POLCOMS which is an even earlier version of AMM.
The revised text makes this link so that the reader can understand what AMM is and how CO4, CO5, POLCOMS are all linked to AMM .
NWS and AMM are similar but not quite the same, the AMM is the modelling system and NWS is the geographical region.

2) Make clear the connection between AMM and CO5
(as above)

3) p2 l6-8 p2 l17-18 are the same:
L6-8: In this paper the subject of interest is what we label the standard Coastal Ocean configuration version 5 (CO5).
L17-18 Here we describe the non-assimilating CO5 control hindcast that provides a reference to understand underlying biases and drifts attributable to changes in the physics updates alone.

Reply:
The reviewer suggests replacing L6-8 with that of L17-18 to avoid repetition.
That is clearly sensible, and contains the information required about this paper being about non assimilating part.

4) P2 l14-17 List of changes should include atmospheric fluxes Reply: Revised text changes to:
"Changes include new riverine forcing, updated Baltic boundary conditions, increased vertical resolution, different surface forcing, as well as updating the base NEMO version from 3.2 to 3.4."

5) Section 3: Add sub-section numbers 3.1 – 3.4 at the appropriate places, e.g., p4.l17, p4.l28

Reply: Can amend as suggested below:
"shorter runs detailed for the forecast implementation of CO4 in O'Dea et al. (2012). Here we describe in detail each of the changes and in Section 4 a set of sensitivity experiments explores the impacts of these changes. 3.1 Relative to CO4, which uses the stretching function in Song and Haidvogel (1994), CO5 features both more model levels (increased from 33 to 51) and uses the stretching function as detailed in Siddorn and Furner (2013) for the terrain following coordinate system. We refer to the stretching function in CO4 as SH and that in CO5 as SF. The new stretching function"
"coupling where again consistent air-sea exchange will be important. 3.2 The second significant change between CO4 and CO5 is the data source for riverine input. In CO4 an annual climatology of some 320 European rivers mapped to 165 outflow points on the CO4 grid constitutes the riverine input regardless of the 30 model year (Young and Holt, 2007). "
Other changes The Third : : : 3.3
And The Surface 3.4

6). p5.l1: Add url for E-HYPE data
Reply: Added reference to http://hypeweb.smhi.se/europehype/long-term-means/
http://hypeweb.smhi.se/europehype/about/

7) p6.l13: Add a referral to fig. 11

Reply:
Have inserted a new Figure reference here to show the masking as suggested by the reviewer.
"No attempt is made to model the Danish straits and they are removed from the domain

as seen in the hashed out region of Fig. 11"

8). p7.l5-: : :: Give a reason for why the surface bdry condition was changed.
"The surface boundary condition in CO5 has also changed from CO4"
Reply:

The purpose was to develop long hindcasts for which the NWP data used in CO4 does not exist.
Whilst this is explained later in the text, the reviewer suggestion here is to bring forth that explanation here where the surface bdy is first discussed. This shall be done in the revised text ti aid the reader.
The later description is in 5.4 year sensitivity runs: "The constraint on the start date is the availability of Met Office NWP flux data and the open (double open a typo) boundary conditions used in CO4 which start from November 2006"

9) . p8.l14: "...amphidrome in southern Norway ... better represented" needs to be substantiated.

Reply:
Have added the reference: "M.J. Howarth, D.T. Pugh, Chapter 4 Observations of Tides Over the Continental Shelf of North-West Europe,
In: B. Johns, Editor(s), Elsevier Oceanography Series, Elsevier, 1983,
Volume 35, Pages 135-142, 143, 145, 147-188, ISSN 0422-9894,
ISBN 9780444421531, http://dx.doi.org/10.1016/S0422-9894(08)70502-6.
(http://www.sciencedirect.com/science/article/pii/S0422989408705026) " However, the issue here is the observations are sparse here. The text needs to reflect this. The text is updated to say that whilst the amphidrome coincides better with the above observationally derived charts, the observations are sparse and there is large uncertainty here. It cannot be said that the change is an improvement. and references to improvement in the amphidrome position are removed.

10.) p8.l15-16: "It is found that..." without further explanation means "unsubstantiated claim" in my book. Was a sensitivity test done? Are the CO4 and CO5 bathymetries otherwise identical?
"represented in CO5. It is found that the change in the bathymetry and land sea mask due to the new Baltic boundary condition is the main driver behind the shift in the amphidrome."

Reply:
The reviewer is correct, the text should have pointed out that the bathymetry for CO4 and CO5 is the same except for the new masked out region and the text is amended to include this detail. Two CO5 like runs where used to check it effect also. This should have been reported in the paper and is included in the revised paper.

11) p12.l15-18: Is this meant as an explanation for why the bottom profiles in fig. 7 look different? If so, say so.
"The location of the transects are chosen to intersect 15 regions of particularly large bias. Note that these comparisons use the CMEMS POLCOMS dataset, which was interpolated onto standard depth levels from the native POLCOMS grid which uses 40 s-levels in the vertical (Holt et al., 2012). The interpolated POLCOMS data is particularly coarse at depth which is reflected in the step like nature of the POLCOMS bias plots at depth".

Reply:
I had thought that was covered by the follow-on sentence but this should be made clearer, being more explicit e.g. ".. step like nature of the POLCOMS bias plots at depth, which accounts for the differences in the bottom profiles in Fig 7"

12) P18.Figure 11: Add "The figure shows CNTL minus S30_1. Grey area in Kattegat indicates interface to Baltic NSBS model data."
"Figure 11. Comparison of mean salinity at 5 m between CO5 with 51 vertical levels using the Siddorn and Furner stretching function (CNTL) and 33 vertical levels using the Song and Haidvogel stretching function (S30_1)."

Reply: This is added and is relevant to the reviewer's suggestion in 7) so that the reader can clearly see the update to the model domain with respect to the changed Baltic bdy.

13) p19.l5: should be "slightly less saline" if the Figure 12 caption is correct.
"The IoW boundary results in a slightly more saline SSS over in the Norwegian trench."

Reply:
The caption actually has part C incorrect and needs to be amended,
Figure 12. Comparing SSS using climatological river and Baltic inputs against EHYPE
rivers and IoW Baltic. Panel (a) 33 SH levels with E-HYPE rivers and IoW
Baltic (S30_1) Vs EN4. Panel (b) 33 SH levels with climatological rivers and climatological
Baltic (S30_2) Vs EN4. Panel (c) E-HYPE rivers minus climatological rivers
(S30_4-S30_2). Panel (d) IoW Baltic - climatological Baltic (S30_4-S30_1).
That should have been S30_3-S30_1 as in the plot title
With regards to the caption for Fig 12 (d) I have double checked the data and scripts
and the caption is incorrect also. It should have read Climatological rivers (S30_4) -
EHYPE rivers (S30_1) . The S30_x labels were correct but not the figure text. However,
the text in the body remains intact. That is the IoW Baltic data results in a slightly more
saline SSS.

14) p13.l13: to me "near bed" means right at the bottom, but Figure 13a shows that the
PDWL scheme reduces bias over more than half the water column. Reformulate.
P19.. "Figure 13(a) compares each run on shelf in regions of seasonal stratification.
Using the PDWL light scheme has three effects; it increases the warm surface temperature
bias, it increases the mid depth cold bias and it reduces the near bed bias.
"

Reply:
The term "near Bed" was incorrectly used here as it comes from information that can't
be ascertained from Fig13 alone. The bathymetry is quite variable over the area considered.
Ranging over the 40-100m depth range shown in the aggregated profile plot
of Fig 13. Thus the bed is often at 40m. However, this is irrelevant in terms of light
schemes and the reviewer is correct to request a change to the text.
e.g. " From 40m to the sea bed" would be a more accurate description"

15) p25.l11: same as previous as above.

16) p22.l21-23: Give a reference for the cold/fresh bias in ORCA025.

Reply:
There Reanalysis inter comparison papers are global in nature and it is difficult to see
from the plots the small patch of the North Atlantic.
Instead we can compare the GLOSEA data against WOA data as was done with CO5
to show the biases in this region and reword to reference the additional figure attached.

17). p22.l25-30: There appear to be two differences at play here: different sources
(Met Office NWP and ERAI) and how they are applied (as direct fluxes or via CORE
bulk formulae). "ERAI fluxes" is a bit misleading; use "ERAI-derived fluxes" or similar
in stead. Is data availability the reason for doing this sensitivity test?

Reply:
Yes, data availability is the issue here. The NWP fluxes do not extend backwards in
time as far as the ERAI fluxes. The paper shall be reworded to ERAI-derived fluxes as
suggested by the reviewer.

18). p22-24 Section 5.42: Why not show the SST/SSS biases in the two runs in Figure
15? I would imagine the discussion would be easier.

Reply:
Bias plots for the short sensitivity runs were avoided in the paper as it was felt that it
may be misleading with respect to the longer 30 year sensitivity experiments. It was
felt that only the relative impact of the shorter experiments to each other should be
used, as such a short period could be anomalously cold/warm and lead to incorrect
conclusions. It does make the discussion more difficult but avoids the issue of short
term anomolous bias plots.
However, if the review feels strongly about this issue, these can be added and the text
modified accordingly.

19). p23.l1-p24.l2 and p26.l2-7: This is interesting and could be substantiated more.
What is the actual impact on the circulation? "Significant impacts" is not very useful.
P23 L1"However, it should also be noted that because direct fluxes use the Haney

correction the resulting model SST is indirectly relaxed to the prescribed SST in a hindcast simulation.": : :
Relevant lines for clarity:
P24 l2-7: "The differing resolution of the surface forcing and the use of absolute instead of relative wind stress is thus likely to play a role in the different sensitivity experiments. In Fig 15(b) it is shown that the ERAI forced experiment is slightly more saline on shelf but significantly fresher in the Skagerrak and the Norwegian Trench mirroring the SST differences here. The difference in the shear stress modifies both the transport of the surface fresh layer 5 out of the Skagerrak and the transport of relatively saline water from the North Sea into the Skagerrak. The difference in relative and absolute winds are significant also along the shelf break from the Shetlands northwards."
Discussion: P26 l6-7: "The combined difference of absolute versus relative winds and differing details in the fluxes combine to have significant impacts in local regions such as the Skagerrak."

Reply:
The reviewer is correct, the discussion here is too blunt and should restate how the salinity is altered (and better still by how much) Panels are included
to show the difference in mean surface currents, SSS and SST to further illustrate the point.

20). P24.l22-24: Needs some explaining, as mentioned above
"Overall the CO5 tides are of comparable quality to CO4. The reference density of 1035 kg m3 used in the control run slightly degraded the tidal predictions. The position of the degenerate amphidrome in southern Norway is slightly improved in CO5 mainly due to a slight change in the land sea mask originating from a change in the Baltic boundary condition."

Reply:
Here we cannot state the amphidrome has improved only that it has changed and that it coincided with poorly constrained observational based estimates of the amphidrome.
21). P24.l29-31: CO5's SSS bias in the German Bight is also affected by E-HYPE.
P24 l29-31 "As in the SST, CO5 has a similar pattern of fresh bias in the near surface salinity from Iceland to the FSC as well as a large fresh bias in the German Bight and a dipole of surface salinity bias along the Norwegian Trench that suggests insufficient lateral mixing. POLCOMS in contrast is slightly too saline in the German Bight and uniformly too saline at the surface along the Norwegian Trench."
Reply: Here we insert as the reviewer suggests the issue with respect to EHYPE data and the German Bight.

General comments: "I also suggest adding a table with the relevant features of the 3 models: POLCOMS, CO4, CO5; it would make the similarities and differences clearer in the discussions in Section 5."
This is very worthwhile suggestion and is added to the revised text to aid the reader.

The author is also grateful for the technical corrections 1-7 and the revised text is address these issues.

Part 2 : List of relevant Changes (page numbers and line numbers correspond to the difftex version of the manuscript)

Part A) Technical points

1-3)Picked up various is/are issues and plural usage of 'data', replaced intercompared with compared replaced

5)"However" with on the other hand Now Page 10 Line 11-12

6) p18.l8: "CO5 and may be" → "CO5, which may be" now changed at Page 19 Line 14

7) p21.l5: "...the 30 year experiments (Section 5.3)" now changed at Page 23 Line 4

Significant other changes to address reviewers comments:

1) Elaborate on AMM term Page 1 Line 22 and Page 2 Lines 1-2
2) Removed repeated definitions of Co5 and move to Page 2 Line 8
3) Made explicit that Co4 was also based on AMM7 Page 2 Line 18
4) Added effect of forcing to list of changes Page 3 line 1
5) Added details on vertical coordinate being z*sigma Page 3 line 23
6) Added in subections 3.1 3.2 3.3 3.4
7) Added Ehype data portal Page 6 Line 8
8) Added new figure to show difference of Ehype regionally Figure 4, Page 7
9) Added reference to hashed out region in Baltic in CO5 Page7 Line 9 also added in caption of Figure 12
10) Justified why we used ERAI and not NWP for Longer CO5 runs  Page 8 Line 7
11) Added Table 1 with list of relevant changes in configurations and referenced it on Page 8 line 19
12) Added reference to Observed tides and noted that while CO5 looks more similar to these references it cannot be said it is improved due to data sparsity in the observations. Page 0 Lines 20-28
13) Added that the z interpolation of POLCOMS data accounts for some of the differences in the plots at depth: P13 Line 30
14) Corrected caption labels on Figure 13 Page 20
15) Added Transport analysis of CO5 with new figure Figure 14  Page 21 and added section that says it appears to be sensible so adding weight to the hypothesis that the E-HYPE rivers have too much fresh water.  Page 20 Line 10 – Page 21 Line 3
16) Changed at the bed to 40m to the bed to better reflect the Figure. Page 21 Line 13
17) Added new Figure 17 to show the bias in the GLOSEA5 data that in turns leads to bais in CO5 Page 24. And make reference to it on Page 24 Line   18 – Page 25 Line 4
18) Amended Figure 19 to include the effects on SST SSS and U.V surface current from changing between direct and relative winds. Referenced Page 25 Line 21.
19) Added that Ehype is leads to bias in CO5 Salinity in German Bight Page 27 Line 21

[revised manuscript text omitted]